# Dynamic spatiotemporal features in action recognition: a multimodal study
Qiuhan Jin[1,2,3], Ding Cui [ORCID][1,2,3] ✉ & Koen Nelissen [ORCID][1,2] ✉

Recognizing and distinguishing actions is a complex cognitive process that relies on integrating various spatiotemporal information. However, the specific contributions of spatial and temporal features to action recognition remain unclear. To address this gap, we conducted fMRI recordings in monkeys as they observed videos of grasping, touching, and reaching actions. Using multivariate pattern analysis (MVPA), we identified distinct action representation patterns across the brain, with most regions of the action observation network (AON) exhibiting a grasping-dominant pattern. This neural representation was consistent with the monkeys' behavioral differentiation of these actions in subsequent categorization tasks. Building on computer vision approaches, we systematically extracted dynamic spatial and temporal features from action videos, capturing evolution of feature information over time, and compared these features with the monkeys' behavioral performance. Our results demonstrate that these features are utilized across a hierarchy and selectively correlate with behavior, reflecting a complex interplay between feature information and key action components. These findings imply a distributed coding strategy in which diverse spatial and temporal features are selectively integrated to form action representations that facilitate recognition or discrimination. Our study provides empirical evidence for current action recognition models and introduces advanced computational tools for analyzing high-dimensional and multimodal data.

When we observe someone grabbing an apple or kicking a ball, we effortlessly recognize and distinguish between these actions. However, this seemingly simple task is computationally demanding, as actions are highly dynamic and multidimensional, involving constant changes in spatial and temporal aspects. Our study aims to address two fundamental questions in action cognition: To what extent do brain representations of action reflect action recognition? How does spatiotemporal information contribute to action recognition and the corresponding brain representations?

Computational models suggest that the spatiotemporal information of actions can be decomposed into specific features, such as effector and object shape, spatial relations, relative motion, and temporal sequence. The hierarchical integration of these features facilitates robust recognition of goal-directed actions from naturalistic video stimuli[1–4]. Electrophysiological and neuroimaging studies have identified brain regions that encode spatial-temporal features, including the superior temporal sulcus (STS), parietal cortex, and frontal areas, collectively known as the action observation network (AON)[2,4–16]. While some studies have examined how these features relate to action processing behaviorally, research remains limited. Existing work suggests that shape, spatial relations, and motion cues effectively influence action discrimination[1,15,17–20].

Traditional studies often examine these features in isolation or indirectly link them to action stimuli, making it difficult to determine how multiple features might simultaneously and directly support action perception. For instance, some studies manipulate stimulus properties (e.g., object shape or motion trajectory) independently across different action stimulus sets, preventing an integrated assessment of their contributions. Others use dynamic dot patterns to mimic naturalistic movement sequences, implying the role of sequential patterns in the naturalistic action indirectly. In the present study, we extract multiple features from each action stimulus and directly relate them to respective classification behavior and/or brain representations, allowing us to evaluate their integrative contributions to action processing.

Despite its importance, the dynamic structure of actions remains underexplored. Previous studies have primarily employed time-locked analyses, revealing a temporal hierarchy where different levels of features are sequentially recruited to anticipate and process action events[16,19]. However, such evidence is scarce. Here, we define dynamics as the temporal evolution of feature information. Both spatial and temporal features (e.g., motion, velocity) evolve continuously, varying moment to moment throughout an action sequence. We incorporate the global influence of dynamics by

[1]Laboratory for Neuro- and Psychophysiology, Department of Neurosciences, KU Leuven, Leuven, Belgium. [2]Leuven Brain Institute, KU Leuven, Leuven, Belgium. [3]These authors contributed equally: Qiuhan Jin, Ding Cui. ✉e-mail: ding.cui@kuleuven.be; koen.nelissen@kuleuven.be

holistically analyzing features across the entire action video (frame × time), thereby capturing the evolution of feature information over time and revealing dynamic features.

The development of two-stream convolutional neural networks (CNNs) and other computer vision tools enables us to construct and extract hierarchically structured dynamic features from action videos for our study[21–25]. However, these dynamic spatial and temporal features exist as high-dimensional numerical arrays, posing significant computational challenges for direct comparison. We adapted point cloud distance estimation techniques, specifically chamfer distance, from the field of computer vision[26–28]. The chamfer distance metric quantifies the (dis)similarity between two sets of points in a multi-dimensional space—in our case, high-dimensional feature arrays. By applying this method, we can efficiently compare multidimensional feature representations and further integrate them with behavior and brain data, which exist at different scales and dimensionalities. Our framework thus provides a computationally efficient solution for multimodal data comparison.

Building on advances in computer vision, we recorded fMRI activity in monkeys as they passively observed videos of three types of manual actions (grasping, touching, and reaching). We selected this trio to implement a graded contact/closure continuum of hand–object interaction, i.e., no contact → contact → enclosure, while holding actor, object, viewpoint, background, lighting, and approach kinematics constant. We then analyzed how these action categories were represented in the brain using multivariate pattern analysis (MVPA). Additionally, we conducted behavioral tests to assess how well brain representations aligned with the monkeys' ability to discriminate between the three types of observed actions. Following this, we compared the monkey's performance with that of the two-stream CNNs to explore potential computational strategies employed by monkeys. We next extracted a wide range of dynamic spatial and temporal features from action videos. These features spanned multiple levels of abstraction, from pixel-level details to high-level motion and shape representations and complex feature maps from deep CNN layers. We then correlated the monkeys' behavioral performance with these dynamic features to assess their contributions to action discrimination. By integrating brain activity, behavioral outcomes, and dynamic spatial-temporal features, our study clarifies how the spatiotemporal visual cues support action classification and brain representation, providing insights into the interplay between sensory and cognitive processes in action perception.

## Results

### Action representation in the brain
We first examined the whole-brain representation of three distinct actions—grasping, touching, and reaching (Fig. 1A)—which share overlapping features but also exhibit key differences. All three actions involved a monkey model directing movement toward a centrally positioned object using either its hand or tail. Both grasping and touching require the effector (hand or tail) to make contact with the object, but only in grasping does the effector fully grasp the object. In touching and reaching, the effector performs similar reaching movements toward the object, but only touching involves contact, while reaching does not.

During fMRI recordings, three monkeys passively observed videos of these actions. A univariate whole-brain analysis revealed activation across early visual areas, STS, parietal, premotor, and prefrontal regions for all action types, with varying activation levels across different regions (Fig. 1A). Due to the asymmetry in the visual field presentation (primarily on the right side), we observed greater fMRI responses in the contralateral (left) hemisphere. However, both hemispheres exhibited similar activation patterns. Early visual areas displayed a retinotopic response pattern, with the greatest activation for grasping, followed by touching, and the least for reaching, compared to static controls. This gradient of activation corresponded to the final position of the effector relative to the object in the visual field. Additionally, grasping elicited more extensive responses in the AON (including STS, parietal, premotor, and prefrontal regions) than touching or reaching, which showed similar levels of activation.

Next, we applied region of interest (ROI)-based MVPA to evaluate how distinctly different brain regions represent these observed actions in terms of action categories. ROIs were selected based on both the fMRI activation patterns described above and previous monkey fMRI action observation studies[10,29–34]. Binary decoding analysis was conducted for pairs of action categories: grasping vs. touching, grasping vs. reaching, and touching vs. reaching. Figure 1B illustrates the average decoding accuracy (heatmaps) across the three monkeys and significantly higher decoding (asterisks) compared to chance level (50%) for each monkey per ROI (detailed results of individual monkeys are shown in Supplementary Table 1). We determined that two action categories were distinguishable if significant decoding was observed in two or more subjects ($p < 0.05$, with or without FDR correction). Based on these results, we identified five distinct action representation patterns: G-T-R, G-TR, G-R, T-R, and GTR (Fig. 1C; details in "Methods"—*Multivariate ROI-based decoding*).

In early visual areas (V1-V4) in both hemispheres, all action categories were represented distinctly. The left area MT and face patch ML in the lower bank of the posterior STS, along with left AIP in the parietal cortex, exhibited full differentiation of the three action categories, corresponding to the G-T-R pattern.

The G-TR pattern, where grasping is distinct from touching and reaching, but touching and reaching are not significantly different, was predominantly observed across several regions of the AON. These regions include MST, FST in the lower bank of left STS; body patch MSB and face patch AL in the middle-anterior sector of left STS; MT, FST, body patch MSB, ASB, face patch ML, AL, STPm in the upper bank, and TEr in the anterior sector of right STS. In the parietal-frontal part of the AON, these ROIs involve the left PFG, premotor F5, and prefrontal 45B on both hemispheres, and 46 v on the right side of the prefrontal cortex.

Some ROIs, such as left ASB and STPa in the anterior upper-bank of STS, along with right AIP and S2 in the parietal cortex, and left 46 v in the prefrontal cortex, exhibited the G-R pattern. This pattern indicates a distinct representation between grasping and reaching, but not between grasping vs. touching or touching vs. reaching. Notably, the left S2 showed a T-R pattern, meaning that it distinguished between touching and reaching, but not between grasping and the other two actions. Together, S2 exhibited somatosensory distinctions, as the G-R pattern on the right hemisphere and the T-R pattern on the left both reflected effector-object contact (grasping and touching), which is absent in reaching.

Finally, in some ROIs—such as upper-bank STS (left STPm, right STPa), left TEr, and parietal S1—no differentiation between the action categories was observed, resulting in the GTR pattern.

### Behavioral discrimination aligns with brain representation in the AON
Many previous studies on monkeys have inferred action-related neural activity under the assumption that monkeys recognize or comprehend observed actions, often without direct behavioral evidence. This raises questions about the extent to which neural activity truly reflects the monkeys' perception or their ability to behaviorally discriminate between different actions. Here, we conducted a behavioral discrimination experiment to assess how well the monkeys' ability to differentiate observed actions aligns with the brain representation patterns identified in the fMRI experiment.

We trained two monkeys, who had participated in the fMRI experiment, on a three-alternative forced-choice action categorization task[29,35]. Using a small set of example videos, the monkeys learned to categorize grasping, touching, and reaching actions by making saccades to corresponding targets (Fig. 2A). We then performed generalization tests with untrained videos, including those used in the fMRI experiment, to assess the monkeys' ability to discriminate between the observed actions.

For the action videos that the monkeys passively observed during fMRI recording, both monkeys successfully categorized grasping actions but struggled to differentiate between touching and reaching as they mostly selected the same target for both categories (Fig. 2B; Supplementary Fig. 1A).

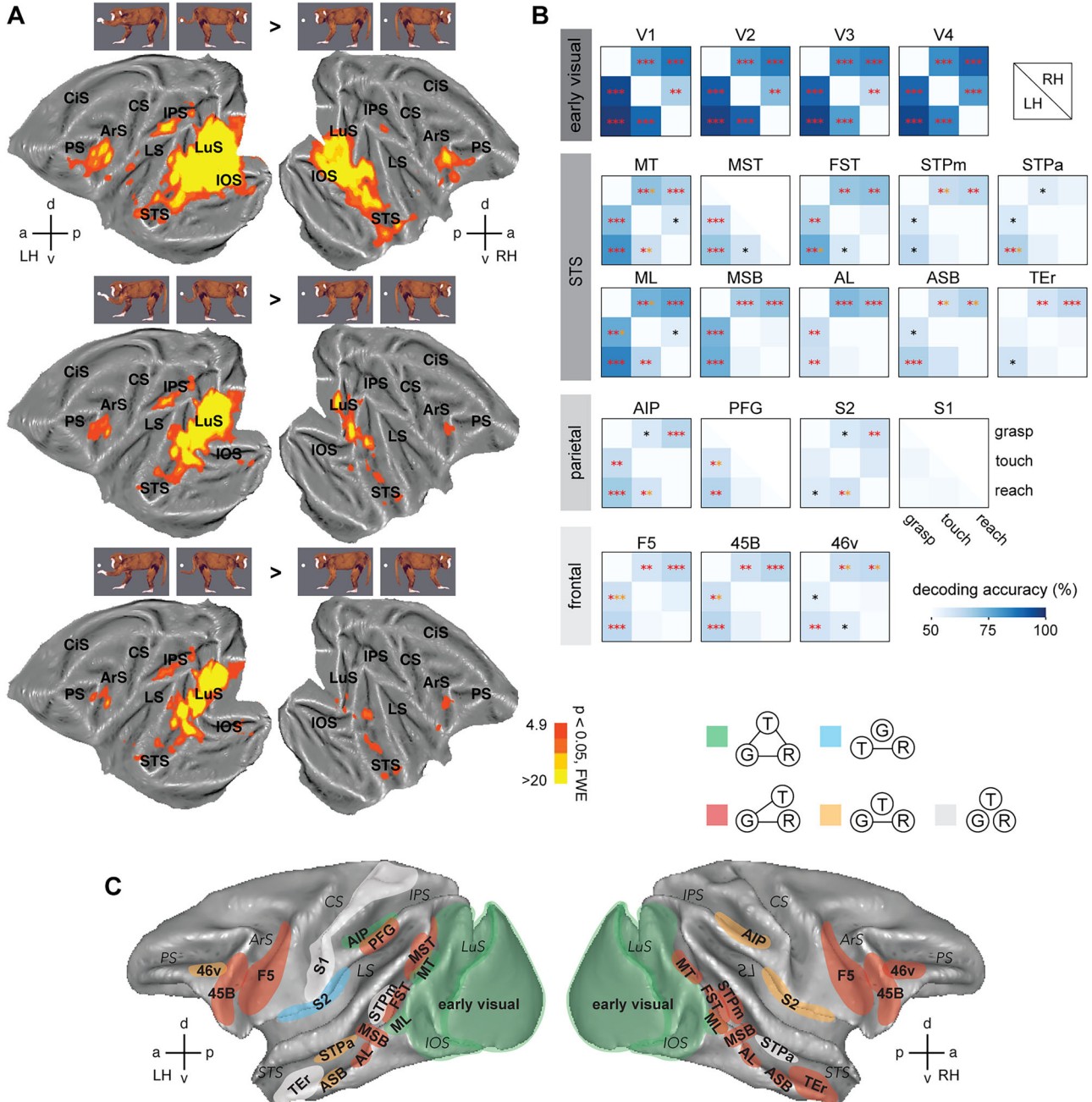

**Fig. 1 | Whole-brain fMRI responses and ROI-based MVPA results. A** Univariate group (n = 3) brain activations (rendered on flattened M12 template brain) for grasping (upper row), touching (middle row), and reaching (bottom row) action observation versus static controls in left and right hemispheres. **B** MVPA decoding of action types in ROIs from early visual, STS, parietal, and frontal regions of both hemispheres. Each heatmap shows averaged decoding accuracies of the three monkeys for pair-wise decoding of the three types of actions (grasping vs touching, grasping vs reaching, touching vs reaching) in the given ROI on the left (lower left) and right (upper right) hemisphere. Each asterisk indicates significant (p < 0.05) decoding above chance level (50%) for one monkey per decoding. For display purposes, for each pair-wise decoding, the asterisks were assigned with redish colors if there are two or all three subjects revealed significant decoding; red color indicates

significant decoding with FDR correction, while orange color indicates significant decoding without FDR correction; black color indicates only one monkey revealed significant decoding irrelevant of FDR correction. **C** Summary of action representation patterns in the ROIs. Each color indicates one of five representation patterns: G-T-R, green; G-TR, red; G-R, orange; GT-R, light blue; GTR, light gray. Each ROI was color-coded based on its representation pattern, which was summarized from the decoding results as shown in (**B**). For details, see Methods—*Multivariate ROI-based decoding*. LuS lunate sulcus, IOS inferior occipital sulcus, IPS intraparietal sulcus, STS superior temporal sulcus, LS lateral sulcus, CS central sulcus, CiS cingulate sulcus, PS principal sulcus, ArS arcuate sulcus. LH left hemisphere, RH right hemisphere.

This behavior revealed a G-TR pattern of categorization, consistent with the brain representations observed in the AON, including the lower-bank STS, parietal PFG, premotor F5, and prefrontal 45B and 46v regions.

Additionally, we tested the monkeys with untrained video sets that varied in action-related features such as viewpoint, object properties, and

effector type (Supplementary Fig. 1A). Across both monkeys, the majority of tests (~56%) exhibited the G-TR pattern, ~33% revealed a G-T-R pattern where all action types were successfully discriminated, and ~11% showed no differentiation, with the same target selected for all categories (Fig. 1C; Supplementary Fig. 1A).

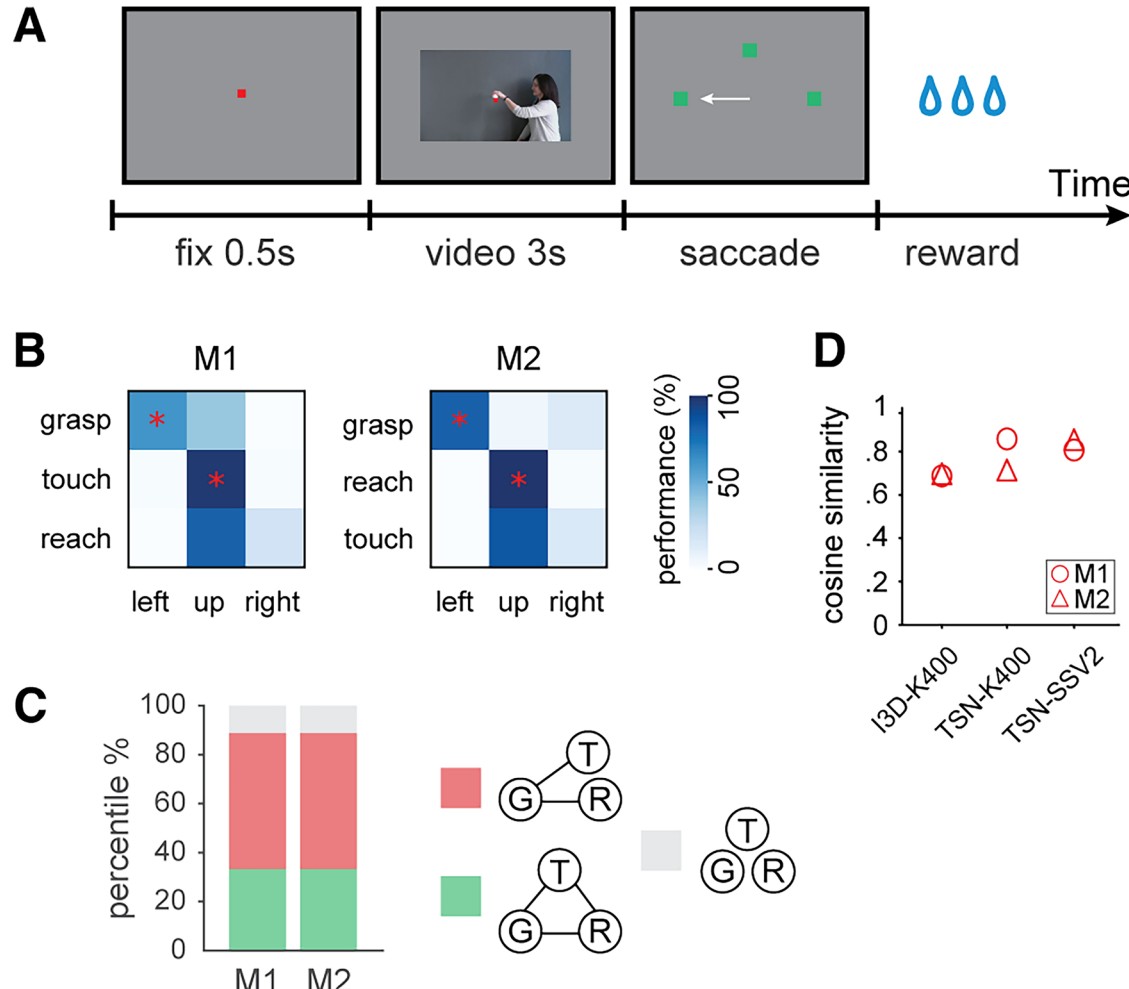

**Fig. 2 | Behavior results of action categorization experiment. A** Example trial of the forced-choice three-alternative categorization task. A trial started with fixation of a centrally positioned red fixation dot for 500 ms, followed by fixation of a 3 s video (with the red fixation dot superimposed at the level of the object) displaying a human actor performing a hand grasp, hand touch or hand reach action. Afterwards, the red fixation dot and the video disappeared, while three green targets were presented simultaneously and the monkeys were required to make a saccade to one of the green targets to receive a liquid reward. The white arrow is only shown for illustration purpose here and was not shown during the actual action categorization task. **B** Behavioral results for generalization tests with the animated monkey action videos (Fig. 1A). The diagonal of a color-coded confusion matrix represents the

categorization accuracies for the corresponding conditions, and the rest indicates the error rate of incorrect selections of the other two targets. "Left", "up", and "right" indicate the location of the corresponding target of action categories in the categorization task. M1 monkey M1, M2 monkey M2. Asterisks indicate significant ($p < 0.05$, binomial test) generalization compared to the chance level (33.33%). **C** Summary of the performance patterns for all the generalization tests. Each color indicates one of three categorization patterns: G-T-R, green; G-TR, red; GTR, light gray. For details, see "Methods" *Analysis of behavioral data for action categorization tasks.* **D** Cosine similarity between categorization performance of the monkeys and prediction scores of CNNs with the same categorization-generalization tasks. See Supplementary Table 2 for detailed results.

In summary, these results indicate that while the monkeys reliably distinguished grasping actions, their ability to discriminate between touching and reaching was limited, as reflected by the predominant G-TR categorization pattern in both behavioral performance and brain representations within the AON.

## Similar action classification of two-stream CNNs and monkeys' behavioral performance

To explore potential computational strategies employed by monkeys for action discrimination, we compared the categorization performance of monkeys with that of two-stream CNN models trained to classify actions based on spatiotemporal information. Specifically, we used an Inflated 3D ConvNet (I3D) and a Temporal Segment Network (TSN) model pre-trained on the Kinetics-400 (K400) dataset, and a TSN model pre-trained on the Something-Something V2 (SSV2) dataset. These models (I3D-K400, TSN-K400, and TSN-SSV2) were designed to rely on the physical characteristics of the stimuli, offering a basis to explore whether similar spatiotemporal patterns might also influence the monkeys' behavioral categorization.

We applied similar training and testing procedures as used in the monkey experiments. This process generated prediction matrices for both the training and generalization test sets (Supplementary Fig. 1B). To quantify the similarity between the monkeys' behavioral performance and the two-stream CNNs' predictions, we conducted cosine similarity analysis between the confusion matrices from the monkeys and the two-stream CNNs. This analysis was performed for training and individual generalization tests (Supplementary Fig. 1C), as well as for pooled matrices across all generalization tests (Fig. 2D).

The cosine similarity analysis revealed a significant correlation (permutation approach, $p < 0.05$) between the categorization patterns of the two-stream CNNs and each monkey (Fig. 2D; Supplementary Table 2).

## Dynamic spatial and temporal features and feature distance

To further evaluate the contribution of spatial-temporal features in the discrimination of observed actions, we developed a comprehensive framework to extract and quantify various stimulus features from the action videos used in the categorization task. These features encompassed multiple

levels of abstraction, ranging from low-level attributes such as luminance and contrast to intermediate spatial and temporal features, as well as high-level abstract structures derived from the deepest layers before the final class output of the two-stream CNNs (I3D-K400, TSN-K400, and TSN-SSV2).

For the estimation of spatial and temporal features, we tailored our approach to capture key action-related characteristics in the videos by applying masks to different body parts (Fig. 3A): hand, hand-and-arm, agent, agent without hand, and agent without hand and arm. Figure 3B illustrates examples of spatial features within the hand-and-arm mask across increasing levels of abstraction, including a histogram of image gradients (HOG), edges, silhouette, silhouette size, and Hu moments of the silhouette. Likewise, Fig. 3C shows examples of temporal features such as optical flow, histogram of oriented optical flow (HOF), motion energy, movement direction, and movement velocity.

A distinctive aspect of our approach to estimating spatial and temporal features is that we accounted for the evolution of each action moment over time, revealing dynamic spatial and temporal features. For instance, as shown in Fig. 3D, when extracting the spatial feature 'edges' from video M or N, we calculated the edges of each individual image frame and pooled this information across all frames along the temporal axis, forming a 3-dimensional dynamic feature. This method was applied to all spatial and temporal features, regardless of their varied dimensionalities, enabling us to preserve the original feature dimensions while capturing their evolution over time. Supplementary Video 1 presents examples of these dynamic features (except for Hu moments) within the "agent" mask.

To quantify the similarity or difference between action videos, we computed the chamfer distance for all features. As illustrated in Fig. 3E, the upper and lower point clouds represent the dynamic feature "edges" extracted from video N and video M, respectively (Fig. 3D). The chamfer distance quantifies the (dis)similarity between these point clouds by computing the average squared Euclidean distance from each point in one cloud (M) to its nearest neighbor in the other cloud (N), and vice versa. These two averaged distances are then summed to obtain the final measure. We applied this pairwise distance computation for each feature between action categories (grasping vs touching, grasping vs reaching, touching vs reaching) across all generalization tests (Supplementary Table 3).

### Dynamic spatial and temporal features reveal distinct correlations with behavior

Finally, we conducted a correlation analysis to explore the relationship between stimulus features and behavioral performance. Since behavior and stimulus features differ in scale and dimensionality, we transformed the monkeys' behavioral performance into "behavior distance", which conceptually mirrors the "feature distance" derived from stimulus features (see "Methods"—*Correlation between behavioral performance and stimulus features*).

Our results revealed distinct correlations between dynamic spatial and temporal features and monkeys' behavioral performance. Results for group-level and individual monkeys are presented in Fig. 4A, B (see Supplementary Table 4 and Supplementary Fig. 3 for details, including permutation-based null), but here we focus on the group-level analysis. Among the dynamic spatial features, both the least abstract feature, HOG, and the most abstract feature, Hu moments, exhibited significant correlations with behavior across all mask types (Fig. 4A). Intermediate-level features, such as edges and silhouette, showed significant correlations only within the hand mask, but not in other mask types. However, silhouette size did not correlate with behavioral performance for any masks. In contrast, dynamic temporal features showed significant correlations primarily for concrete features such as optical flow and HOF. Specifically, optical flow was significantly correlated with behavior in the hand and hand-and-arm masks, while HOF exhibited significant correlations in all masks except for the mask of the agent without hand and arm (Fig. 4B). Overall, these results suggest a complex interplay between monkeys' behavioral discrimination of observed actions and both localized and broader spatial-temporal information.

In addition, low-level features such as luminance and contrast did not show significant correlations with behavioral performance (Fig. 4C). In contrast, high-level abstract features derived from the two-stream CNNs demonstrated a significant correlation with the monkeys' behavior (Fig. 4D).

Finally, layer-wise correlations (Supplementary Fig. 2; Supplementary Table 5; Supplementary Fig. 4) show that: across all three models, the top two layers 5–6 reliably correlate with behavior in both individual monkeys and the group; in TSN-SSV2, significant correlations also appear at intermediate layers 3–4; and in I3D-K400, significant correlations extend across the entire hierarchy, including the lowest layers 1–2.

## Discussion

Using MVPA, our study provides a detailed map of how different types of actions (grasping, touching, reaching) are represented in the macaque brain, revealing distinct representation patterns across various regions. A substantial portion of regions within the AON exhibits a grasping-dominant pattern, which aligns with the monkeys' ability to discriminate between these types of actions. The monkeys' performance closely mirrors that of two-stream CNNs on the same tasks. Furthermore, the monkeys' action discrimination patterns are distinctly correlated with representations of a hierarchy of dynamic spatial and temporal features, which capture the evolution of feature information over time. To achieve this, we introduced a framework that applies advanced computer vision methods to systematically extract multidimensional features from action videos and further compare them with behavioral data, despite differences in scale and dimensionality.

Our study provides empirical evidence addressing the mechanisms underlying action recognition, as proposed by computational models[1–4]. We observed significant correlations between monkeys' performance patterns and that of two-stream CNN models (Fig. 2D). This suggests that monkeys, like artificial neural networks, may process spatial and temporal features hierarchically, integrating various levels of abstraction to recognize actions[21–25]. Supporting this, our findings show that individual spatiotemporal features, which significantly correlated with behavior, span a hierarchy: from concrete features like HOG, edges, optical flow, and HOF, to intermediate-level silhouettes, and up to abstract features like Hu moments (Fig. 4A, B). Similarly, behavior-CNN layer correlations (Supplementary Fig. 2; Supplementary Table 5; Supplementary Fig. 4) show that behavior-relevant information is present throughout the CNN hierarchy, indicating distributed recruitment of spatiotemporal features across depth. Furthermore, we analyzed stimulus features holistically throughout action sequences (frame × time) to capture evolving feature information. This approach aligns with the design principles of two-stream CNN architectures like I3D and TSN[36,37], which utilize 3D convolutions to simultaneously process spatial and temporal information, or model temporal relationships of spatial information to obtain overall action dynamics. Taken together, these findings support the interpretation of convergent solutions under shared task constraints between monkeys and the CNNs in how spatial and temporal information are leveraged during action processing.

Another critical mechanism in action processing may involve the interplay between spatiotemporal features and key action components. For instance, a previous study found that subjects' similarity judgments of actions were independent of the optical flow of the entire action stimulus[19]. Our findings corroborate this by showing an insignificant correlation between action discrimination and optical flow in the whole-agent mask. However, when focusing on effectors (hand and/or arm), optical flow and HOF information within these masks reveal significant correlations with behavior. Additionally, we observed that while behavioral performance is related only to hand edge and hand silhouette, all action components (hand, arm, and body) of HOG and Hu moments significantly contribute to behavior. This diverse involvement of action components in action processing aligns with the findings of Tarhan and Konkle[38], which show that varied configurations of action components correspond to distinct brain representations for respective action categories. Thus, we propose that the

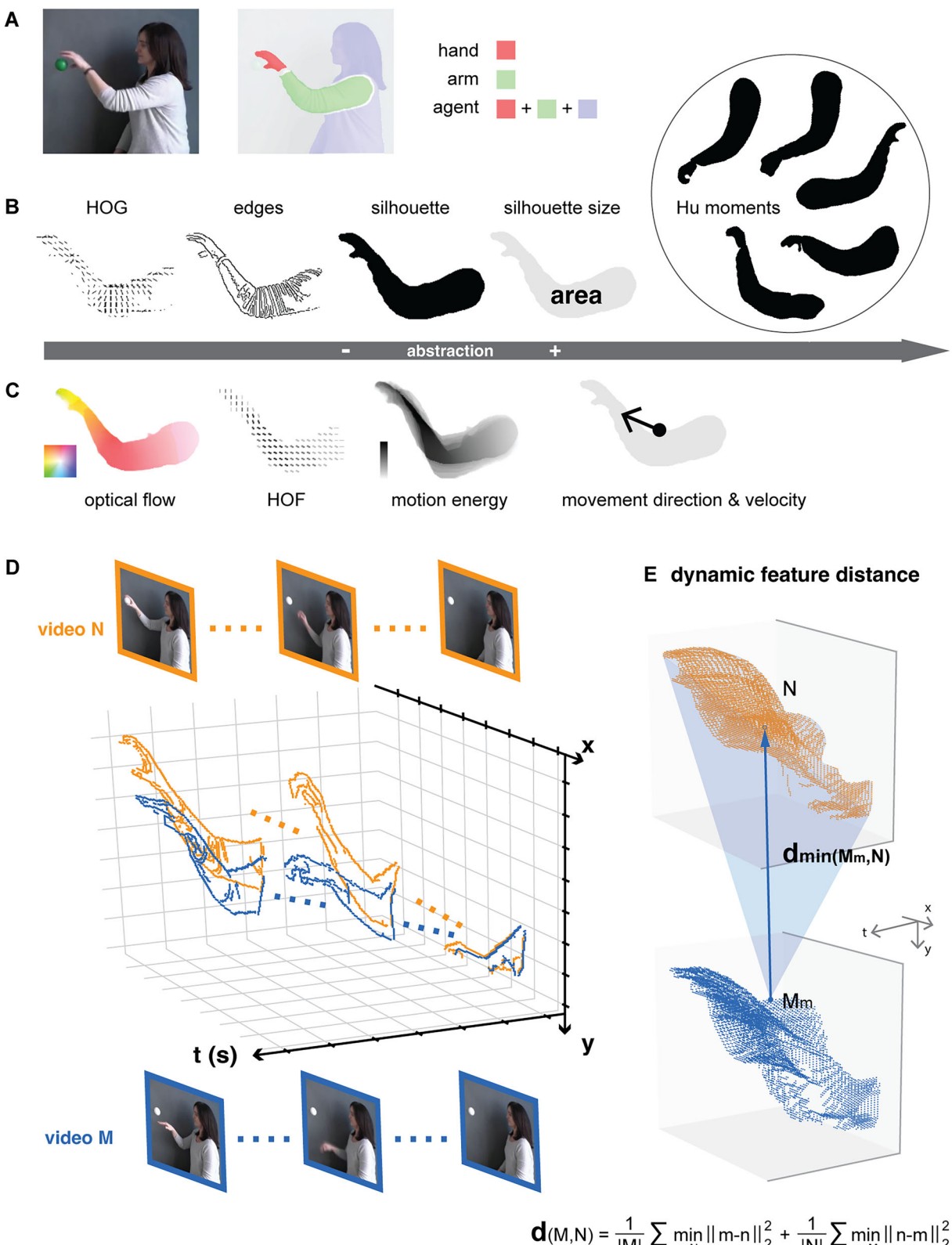

**Fig. 3 | Dynamic spatial and temporal features and dynamic feature distance of the action videos. A** Mask definition of the effectors (hand and arm) and the agent in the action videos. Examples of dynamic spatial (**B**) and temporal (**C**) features within the mask hand-and-arm. Hu moments—a feature that is invariant to transformations such as translation, scaling, rotation, and reflection. Optical flow color wheel: hue indicates direction, saturation indicates magnitude. Motion energy color bar: the darker the color, the higher the motion energy. The dot indicates the center of the masked action frame, the direction of the arrow indicates the movement direction, the size of the arrow indicates the speed of the movement. For details, see "Methods"—*Estimation of dynamic spatial and temporal features*. **D** Examples of a type of dynamic feature (edges) for two action videos showing in the same 3D space. **E** Illustration of computing the dynamic feature distance between two videos using the chamfer distance metric. For details, see "Methods"—*Correlation between behavioral performance and stimulus features*.

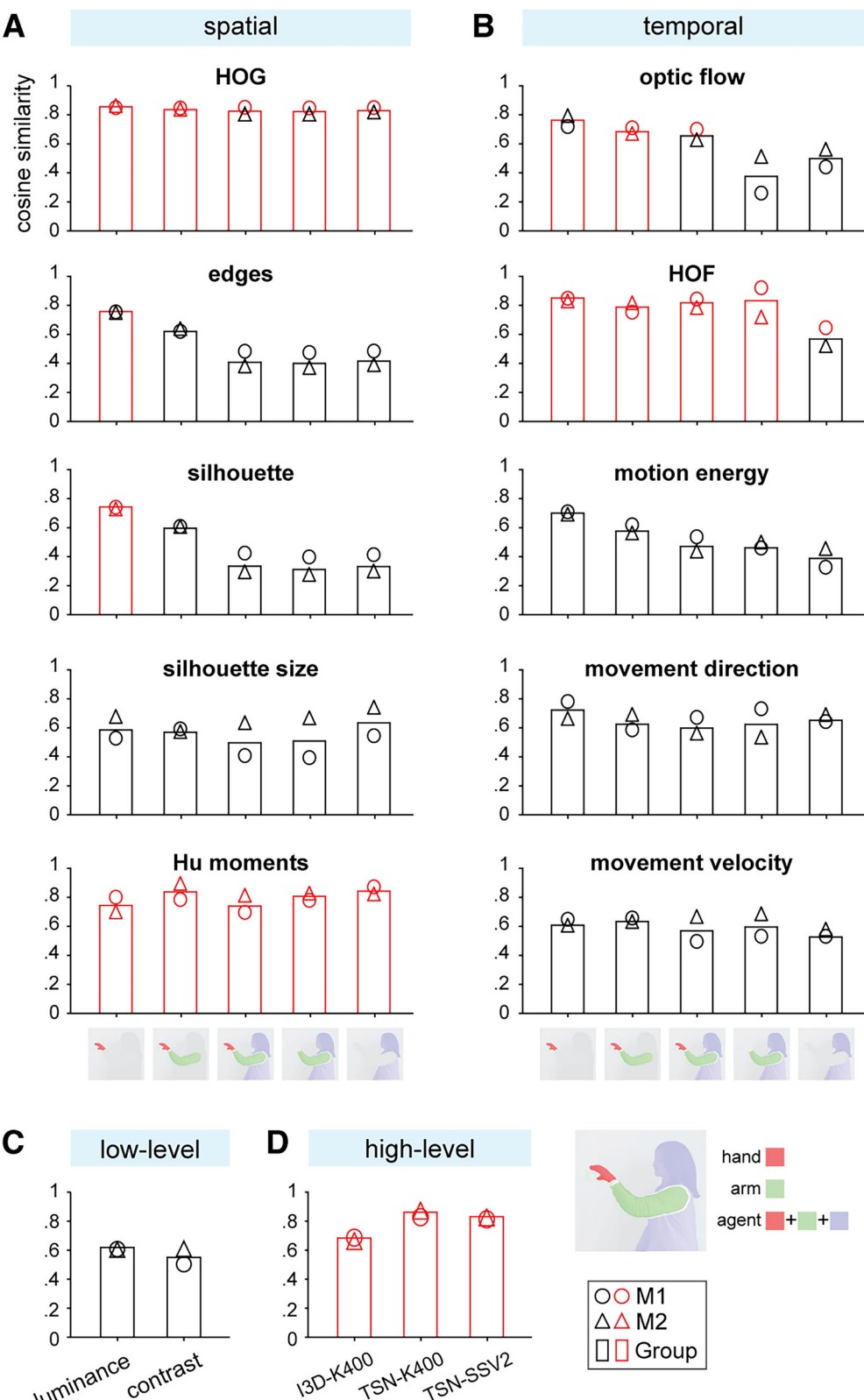

**Fig. 4 | Correlation between stimulus features of the action videos and monkeys' behavioral performance.** Cosine similarity between behavioral performance and **A** dynamic spatial features, **B** dynamic temporal features, **C** low-level features, **D** high-level features derived from the last layers of the two-stream CNNs. See Supplementary Table 4 for detailed results. Red color indicates significant correlation ($p < 0.05$, with a permutation test). M1: monkey M1, M2: monkey M2.

multi-feature integration mechanism also interacts with diverse encoding of key action components to support action recognition.

Furthermore, both previous studies[18,19,38,39] and our findings (Fig. 1B, C; Fig. 4C) indicate that low-level pixel-based features of actions do not sufficiently explain behavioral performance, and they are primarily processed in the early visual cortex. In our results, high decoding accuracies in V1–V4 are expected from sensitivity to retinotopic layout and low-level variations that differ across the three action categories. In contrast, our

https://doi.org/10.1038/s42003-026-09917-z                                                                                                    **Article**

results demonstrate that the overall action representations in the AON closely align with monkeys' ability to discriminate observed actions. Behavioral performance exhibits selective correlations with diverse spatial and temporal features, highlighting the distinct contributions of these features to action representation. Consistent with this view, prior work shows that behaviorally relevant features better track AON representations than generic labels or low-level summaries[40]. Collectively, these findings suggest that the AON employs a distributed coding strategy, selectively integrating multiple visual cues to form action representations that facilitate recognition or discrimination. Specifically, building upon the action recognition model proposed by Fleischer et al.[2], and other previous studies[10,13,15,41–50], our results can be interpreted within the context of the proposed hierarchical organization in the AON (without asserting a processing hierarchy). The posterior STS is primarily tuned to motion-related features such as optical flow and HOF. As visual information progresses along the STS, shape-related spatial features are processed in a hierarchical manner: the posterior STS detects edge orientations and contours (HOG and edges), the middle STS, including face and body patches, encodes spatial arrangements and body part shapes, such as hand silhouettes, and the anterior STS extracts invariant shape representations such as Hu moments. Additionally, parietal regions may either process motion and shape information independently or integrate inputs from STS regions to generate higher-order representations. Finally, diverse spatiotemporal features converge in premotor and prefrontal cortices, facilitating higher-order action perception. Supporting this framework, our results reveal that regions such as F5, 45B, and 46v exhibit uniform representation patterns that closely align with behavior, whereas STS and parietal regions display more diverse representation patterns (Fig. 1C).

Notably, our results also indicate the presence of other action-related features represented within the AON. For instance, beyond optical flow, responses in MT also encode other spatiotemporal features[41], which may explain the observed mixture of representation patterns (G-T-R and G-TR) in MT. Similarly, AIP exhibits a mixed representation pattern akin to MT, which may be because parietal regions process both concrete and abstract action-related features, including 3D shape, motion, movement trajectory, action type, and action goal[51–54]. Interestingly, unlike other AON regions that exhibit behaviorally aligned representations, action representations in S2, but not S1, suggest a somatosensory distinction during action observation. Specifically, S2 dissociates effector-object non-interactive reaching from grasping and touching, reinforcing its role in action-related sensory processing[31,55]. It is important to note that within the AON regions showing alignment with behavioral representations, we cannot rule out the influence of additional action-related features—such as action goals, transitivity, and viewpoint—which were not the focus of our present study but may closely correlate with the behaviorally relevant spatiotemporal features we identified[12,40,56,57]. To clarify the respective contributions and potential interactions between these features, future research should explicitly investigate overlapping and/or dissociable representations of spatiotemporal features, as proposed in our study, alongside other action-related attributes across AON regions.

Our stimulus set samples a graded contact/closure continuum (reaching, touching, grasping) under tightly controlled scene and kinematic factors to maximize interpretability for feature-behavior-brain comparisons. The near-boundary nature of touching vs reaching produces higher confusability than grasping and is informative for our feature-behavior focus because it reveals which dynamic cues are weighted when discriminability is low. While this design isolates diagnostically relevant spatiotemporal information, it may limit inferences about broader action taxonomies. Future work will expand the action space (e.g., transitivity, tool use, social vs nonsocial contexts, viewpoint/object variation) to test whether the patterns we observe generalize across richer repertoires. In parallel, direct feature-brain mapping (e.g., voxel-wise encoding models or condition-rich RSA with a larger, more varied action set) will be a natural next step to fully characterize the representational organization of our dynamic features in the primate brain.

Our study introduces several methods that enhance feature computation, comparison, and multimodal data integration. Traditionally, studies have averaged feature information across video frames or employed techniques like principal component analysis to reduce feature dimensions, aiming to improve computational efficiency. However, these approaches risk losing critical information. In contrast, we conducted a comprehensive analysis by preserving all feature information across all dimensions and structuring the features using a frame-by-time scheme to retain evolving information over time. Furthermore, we adapted the chamfer distance metric from computer vision for direct comparisons between high-dimensional feature arrays. This metric considers contributions from all dimensions of the feature space, aligning with our goal to assess features holistically across entire action sequences. Additionally, its computational efficiency can be significantly enhanced through the use of symbolic matrices[58], addressing challenges associated with high-dimensional feature arrays. Our study also encompasses heterogeneous datasets that vary in scale and dimensionality, including fMRI brain responses, nominal behavioral performance, and feature information from action videos. By applying MVPA, Youden's J statistic (see "Methods"), and the chamfer distance metric, respectively, we transformed these data into a common metric space, representing similarities and dissimilarities across actions. This approach facilitated both qualitative and quantitative comparisons between these data types, enabling the integration of diverse evidence. Overall, our methods are broadly applicable to studies utilizing naturalistic videos, requiring high-dimensional feature analysis, and/or involving multimodal data types.

## Methods
### Subjects
Three male rhesus monkeys (Macaca mulatta, 5–7 kg, 5–7 years old; M1, M2, M3) participated in the present study. All experimental procedures and animal care followed the national and European guidelines and were approved by the animal ethical committee of KU Leuven.

### Action observation task
The three monkey subjects (M1, M2, M3) were trained to sit in a sphinx position in a custom-made MRI-compatible chair. They were required to maintain fixation within a 2 × 2° window centered on a red dot (0.2 × 0.2°) displayed in the middle of a computer screen positioned 57 cm from their eyes. Eye position was monitored at 120 Hz through pupil position and corneal reflection (ISCAN Inc.). Rewards (juice drops) were delivered continuously if monkeys maintained fixation on the red dot, with or without action videos displayed in the background.

We created videos of a 3D animated monkey model using the open-source Blender software (https://www.blender.org/), which depicted three types of actions (grasping, touching, or reaching) using two effectors (hand or tail)[29]. The grasping action involved the monkey model reaching for and grasping a centrally positioned white ball, either using its left hand or tail. The touching action involved the monkey model reaching for and touching the bottom part of the white ball with its hand or tail. The reaching action displayed the monkey model performing a reaching movement towards the white ball (similar to the touching action), but without making contact with the object. Static images of the monkey model in a standing position, either facing towards or away from the white ball, were used as control conditions for the hand and tail actions, respectively. Figure 1A shows key frames of the action videos and the static images. The white object was positioned at the center of the screen, while the animated monkey model was positioned on the right half of the screen. Each video lasted 6 s.

We selected reaching, touching, and grasping to vary contact and enclosure along a graded continuum while holding approach kinematics and scene layout constant (same actor, object, viewpoint, background, lighting). This controlled manipulation isolates effector–object interaction while minimizing unrelated variance, enabling targeted analysis of dynamic features. The compact action set also ensured monkey training and fMRI sampling efficient, with held-out video samples used to assess generalization (see *Action categorization task*).

## fMRI data acquisition

FMRI data were acquired with a 3 Tesla full body scanner (Siemens, Prisma fit) using a gradient-echo T2 $*$ -weighted echo-planar imaging sequence of 40 horizontal slices (time repetition [TR], 2 s; time echo [TE], 17 ms; $1.25 \times 1.25 \times 1.25$ mm3 isotropic voxels) with a custom-built 8-channel phased-array receive coil, and a saddle-shaped, radial transmit-only surface coil. Before each scanning session, an iron contrast agent (Molday ION, BioPAL) was injected intravenously (9–12 mg/kg) to enhance the spatial selectivity of the MR signal changes and, accordingly, improve signal-to-noise ratio[59]. We inverted the sign of all beta values to account for the difference between iron contrast agent CBV and BOLD activation maps (i.e., increased brain activation produces a decrease in MR signal in iron contrast agent CBV maps).

We conducted fMRI recordings with all three monkey subjects during the action observation task. A block design was utilized, consisting of alternating blocks of a total of nine conditions: fixation only, observing hand-grasp, tail-grasp, hand-touch, tail-touch, hand-reach, tail-reach, hand-static, tail-static. We generated 10 pseudo-random orders of the alternating blocks of all conditions, and each was used and repeated once during a single run. Each block consisted of 15 volumes (30 s). A complete run totaled 9 min and 10 s, during which 275 whole-brain volumes were acquired, including 5 dummy volumes at the beginning. For monkeys M1, M2, and M3, respectively, 39, 36, and 35 runs were collected, while 5, 1, and 1 run was discarded for each respective subject due to poor fixation performance (<90%). Note that the fMRI data from M1 and M2 were previously utilized in a separate study[29], which employed different analytical methodologies.

## Analysis of eye movement data from fMRI experiments

To quantify the monkeys' eye movement behavior during the fMRI recordings, we calculated the overall percentage of fixation and the average number of saccades per minute during observation of each of the six types of action videos per fMRI run. A one-way repeated measurements ANOVA was performed and revealed no significant differences in fixation behavior among the action conditions: % fixation: M1 = 95.94%, $p = 0.63$; M2 = 95.42%, $p = 0.79$; M3 = 96.65%, $p = 0.17$; # sacc/min: M1 = 8.22, $p = 0.6$; M2 = 6.01, $p = 0.66$; M3 = 7.02, $p = 0.59$.

## Univariate whole-brain-based analysis

fMRI images were first realigned using Statistical Parametric Mapping (SPM12) software, and then registered to a template anatomy (M12)[34] through non-rigid co-registration (using JIP, http://www.nmr.mgh.harvard.edu/~jbm/jip/). Functional volumes were resliced to 1 mm³ isotropic and smoothed with a 1.5 mm (FWHM) Gaussian kernel using SPM12. A general linear model (GLM) was used for estimating the response amplitude at each voxel (using SPM12) following previously described procedures[59]. Stimulus conditions were presented as a boxcar model convolving with a MION hemodynamic response function[59]. All nine conditions (see fMRI data acquisition) were modeled as regressors of interest. To account for head-motion and eye-movement related artifacts, six regressors corresponding to three rotations and translations along the x, y, and z-axis, and three regressors corresponding to horizontal and vertical components of eye position and pupil diameter were included as covariates of no interest.

In addition to the analysis on individual monkeys, we performed a fixed-effect analysis on the combined data from all three monkeys. The contrasts of each action type versus static controls (e.g., hand-grasp + tail-grasp > hand-static + tail-static) were calculated, and the significance level was set at $p < 0.05$, FWE corrected. For display purposes, SPM T-contrast maps were presented on a flattened 3D image of both template hemispheres (template M12) using Caret software (Fig. 1A).

## Multivariate ROI-based decoding

We selected a series of ROIs to perform MVPA using a Matlab-based Decoding Toolbox[60]. ROIs were defined within the early visual cortex as well as in different portions of the AON (STS, parietal cortex, frontal cortex).

Definition of the ROIs was based upon previous retinotopic mapping studies for early visual ROIs[34] and on several previous studies examining action observation in monkeys for the AON ROIs[10,29–33]. More specifically, the ROIs included the following regions in both hemispheres (Fig. 1B, C): early visual areas V1, V2, V3, and V4 (which serve as input regions for the AON); areas along the posterior to the anterior extent of the STS—MT, MST, FST, STPm, STPa, TEr; parietal regions AIP, PFG, S1, S2; frontal regions F5, 45B, 46 v. To examine in particular voxels yielding visual responses to our action videos, we selected those voxels from these anatomically defined ROIs that yielded responses for the main contrast of all actions vs static controls at the threshold of $p < 0.05$, uncorrected. Moreover, we included functionally defined body and face patches—MSB, ASB, ML, AL—from our previous study[29].

Using a similar procedure for GLM fitting using SPM12 (see above), T-contrast maps for single action categories per run were computed to serve as input for a linear support vector machine (SVM) classifier for the ROI-based MVPA decoding. The contrasts included hand grasp vs hand static, hand touch vs hand static, hand reach vs hand static, tail grasp vs tail static, tail touch vs tail static, tail reach vs tail static. Next, t-value features per ROI per run were generated by grouping conditions based on action types, irrespective of effector types, for pairwise decoding between action types, i.e., hand grasp + tail grasp vs hand touch + hand touch; hand touch + tail touch vs hand reach + tail reach; hand touch + tail touch vs hand reach + tail reach. The t-value features were utilized for both training and testing the classifier.

We applied a leave-one-run-out cross-validation scheme, where at each iteration the extracted t-value features per ROI from all but one run were used to train the classifier, while data from the remaining run was left out for testing until all runs had been tested once. The average classification accuracy of all iterations per ROI served as the decoding result for each testing ROI, respectively. To determine the statistical significance of the classification performance, we applied a permutation test (per ROI) in which the conditions associated with each feature (or label) were randomly shuffled, and a cross-validation classification scheme was applied for each iteration. This procedure was repeated 1000 times, leading to 1000 classification performance values, thus producing a null distribution of classification performance per ROI. Based on the classification performance of the original classification relative to the null distribution, $p$-values for significance were calculated. P-values less than 0.05 were deemed significant, and FDR correction was applied.

We defined the activation patterns of two action types (e.g., grasping vs touching) as dissociable if two or all three subjects showed significant decoding ($p < 0.05$, with and/or without FDR correction) for the given pair of actions. We summarized five action representation patterns (Fig. 1C) for all the ROIs based on the decoding results from the three monkeys. (1) G-T-R: The activation patterns of all three action types are different from each other. (2) G-TR: The activation patterns of grasping are dissociable from touching or reaching, but touching and reaching are not different from each other. (3) G-R: The activation pattern of grasping is different from reaching, while the activation pattern of touching cannot be significantly classified from either grasping or reaching. (4) GT-R: The activation pattern of reaching is dissociable from grasping and touching, while activation patterns of grasping and touching are similar. (5) GTR: Activation patterns of all three actions are not significantly different.

## Action categorization task

Following the fMRI recordings involving action observation tasks, we trained two of the three monkeys (M1, M2) from which fMRI data were collected in a three-alternative forced-choice action categorization task. The monkeys learned to categorize three types of action videos (grasping, touching, and reaching) by making saccades toward corresponding targets (Fig. 2A). Each trial began with fixation of a centrally positioned red dot ($0.2 \times 0.2°$) for 500 ms, followed by a video displaying one of the three actions for 3 s. During the video display, the monkeys were required to maintain fixation until the video disappeared, necessitating 100% fixation.

Afterward, three green squared targets ($0.8 \times 0.8°$), located at $10.5°$ to the left, up, and right of the center of the screen, showed up, replacing the central fixation dot and the video. For monkey M1, grasping, touching, and reaching actions were associated with the left, upper, and right targets, respectively; for monkey M2, the left, upper, and right targets were associated with the grasping, reaching, and touching actions, respectively. The monkeys had to make a saccade towards one of the targets to receive a juice reward. The next trial started automatically once the monkeys made a saccade to one of the targets or if no response was made after 3 s. Trials with no responses or trials that were aborted due to failure to maintain fixation until the end of the video were excluded from data analysis.

Training. In initial training trials, only the correct target corresponding to the specific action category was presented after the video. Subsequently, the monkeys were trained to categorize the three actions by selecting the correct target when all three targets were presented following the video. For initial categorization training, we used three action videos depicting a female actor performing a grasping, touching, or reaching action towards a white ball. Later, 3 additional video sets featuring either a different object color (blue) or a different (male) actor were gradually introduced. In total, 12 videos were randomly presented during the training sessions (3 action categories $\times$ 2 actors $\times$ 2 objects). Both monkeys completed around 400–800 trials per category for each training session, with 35–40 sessions per monkey. Once both monkeys maintained an accuracy above 80% (per action category) for a few consecutive training sessions, we proceeded with generalization tests. Refer to Cui. et al.[29], for more detailed procedures and behavioral performance regarding the training of the action categorization task.

Generalization Tests. We subsequently tested whether the monkeys could generalize the learned categorization rule to untrained novel videos of the three action categories, including the 3D animated monkey videos used in the action observation task. Each generalization testing session contained ~90% of trials depicting familiar, trained videos, and the rest of the trials (~10%) consisted of novel, untrained videos (see Stimuli below). During generalization tests, monkeys were rewarded no matter which target (left, up, or right) was chosen by the monkey during categorization of the untrained videos, to avoid introducing learning effects. In total, generalization tests used nine different sessions with nine sets of action videos (see Stimuli below), including each time one set of 3 novel videos (one for each action category). We performed 6–10 testing runs (15–20 min/run) for every testing session. Each testing run contained in average 50 and 5 trials per category of trained and untrained videos, respectively. Thus, in total, the monkeys were tested with 900–1500 trials with 90–150 novel untrained videos per testing session.

Stimuli. In the present study, we used the same stimuli as in Cui et al.[29]. All details regarding the stimuli are explained in detail in that study and thus, here we only provide a brief description of the stimuli. In addition to the animated 3D monkey videos depicting grasping, touching, and reaching, we recorded the same three types of actions performed by different human actors, including objects with various colors and shapes. Example key frames of the grasping actions used in the generalization tests are shown in supplementary Fig. 1. In total, the videos included four different actors (2 male and 2 female) performing the actions (with their left hand) toward different objects. These objects either consisted of a medium-sized ball (three different colors), a small ball, or a ring. Additionally, one female actor also performed the actions with her right hand. Finally, as all videos using during training and generalization tests depicted actors filmed from their left side, in the generalization tests, we also included a mirrored (reflected along the vertical axis) version of a training video set, depicting the female actor performing the actions from the left side of the object (flipped view). The videos ($13.9 \times 8.4°$) were presented in the center of the screen with the fixation dot overlayed on the object. The action videos depicting the 3D animated monkey model that were used in the generalization tests were resized, and their mean luminance was adjusted to match the size and luminance of the human action videos, which were used in the training and generalization testing.

## Analysis of behavioral data for action categorization tasks

To assess the behavioral performance of the subjects on the action categorization tasks, for each monkey, we analyzed all completed trials from the last three sessions of the training data (performance optimized and stable) and all sessions for each generalization test. For both training and testing data sets, we made confusion matrices (Fig. 2B, supplementary Fig. 1A) by computing the proportion of category choices for each presented category per data set. The diagonal of a confusion matrix represents categorization accuracies for the corresponding conditions, while the remainder indicates the error rate of incorrect selections towards the other two targets. Binomial tests were performed on proportions of correct trials for each action category against the chance level (33.33%). The formula we used is: $z = (\hat{p} - \bar{p}) / \sqrt{\bar{p}(1 - \bar{p}) / n}$ ($\hat{p}$ = proportion correct trials, $\bar{p}$ = chance level, $n$ = total number of trials). $P$-values (one-tail) were correspondingly obtained from the $z$-values. $P$-values less than 0.05 indicate significantly higher performance than chance level. Note that the results for the generalization tests (excluding those for the training datasets) were previously reported in Cui et al.[29]. In the present study, we have included these results in Supplementary Fig. 1 to facilitate comparisons between training and generalization testing of the monkeys' performance (Supplementary Fig. 1A), as well as to compare action classification performance between the monkeys and the two-stream CNNs (see *Action classification of the two-stream CNNs*; Supplementary Fig. 1A, B).

Similar to defining patterns for action representation (see Multivariate ROI-based decoding), we defined performance patterns according to the confusion matrices for the generalization tests. G-T-R: monkeys successfully generalized the three action types, i.e., performance for each category is significantly higher than chance level. G-TR: Only grasping could be generalized into the correct category, but not touching or reaching. GTR: monkeys completely failed the generalization test. During these tests, both monkeys showed a strong bias to the upper target for trials involving novel videos (Fig. 2B, supplementary Fig. 1A). Therefore, to decide for the performance patterns, we considered two factors: the significant generalization of individual action categories, and the selective ratio of the upper target for non-corresponding categories. Specifically, test 1–3 (Supplementary Fig. 1A) in monkey M1 and test 1, 2, 5 in monkey M2 revealed significantly higher performance for all three actions than chance level, indicating a performance pattern of G-T-R. Test 5–9 in monkey M1 and test 4, 6–9 in monkey M2 were assigned to the pattern G-TR. While grasping revealed significant generalization, monkeys selected the upper target in high instances (>50%) for both touching and reaching, indicating an unclear discrimination for the two action types. Finally, the performance patterns for test 4 in monkey M1 and test 3 in monkey M2 were categorized as GTR since the confusion matrices either show a clear biased upper target selection or random target selection for all action types. We then summarized the performance patterns for all the generalization tests by ratio (Fig. 2C).

## Action classification of the two-stream CNNs and the layer features

Two-stream CNNs, inspired by the two-stream hypothesis of the primate visual system, consist of two separate architecture streams: a spatial stream that processes RGB frames to capture appearance (spatial) information, and a temporal stream that processes optical flow to capture motion (temporal) information[21,23]. Convolutional layers extract hierarchical features from input frames. Early layers capture low-level features like edges and textures, while deeper layers reduce the dimensionality of the feature space, distill the most relevant information for action classification, and create a compact, discriminative representation of actions[24,25]. This provides us with a powerful tool to exploit high-level abstract representations of our action videos. We first applied a similar training and testing procedure with the two-stream CNNs as in monkeys to examine their categorization performance and compared it with monkeys' performance. Then we extracted the CNN layer-derived features from each model for subsequent analysis.

We employed three open-source pre-trained two-stream CNN models provided by MMACTION2. The two-stream CNNs are Inflated 3D ConvNets (I3D)[37] and Temporal Segment Networks (TSN)[36], both using 3D convolutions to exploit spatiotemporal information of naturalistic action stimuli. I3D uses inflated 3D convolutions, enabling spatiotemporal integration throughout the hierarchy and capturing fine-grained temporal structure more directly. TSN is designed to model long-range temporal structure by sampling multiple segments and combining them via a consensus function. The pre-trained models are an I3D and a TSN pre-trained on the Kinetics 400 dataset (K400)[61], a dataset of 400 human action classes involving full body movements, and a TSN pre-trained on the Something-Something V2 dataset (SSV2)[62], a collection of labeled video clips that show humans performing basic manipulative actions involving the movement of forelimb effectors. Specifically, we applied the three instances of two-stream CNN models, namely, I3D pre-trained on K400 (I3D-K400), TSN pre-trained on K400 (TNS-K400), and TSN pre-trained on SSV2 (TSN-SSV2). This manipulation allowed us to test whether the performance is robust across different architectures and learned priors, rather than being tied to a single source. All models were fine-tuned on our 12 training action videos (as for monkeys) using a 6-fold cross-validation framework to classify a video into grasping, touching, or reaching action. We saved models after each training epoch and selected models from the first epoch when the learning curve reached the plateau. The models were then applied to predict the action category of the testing videos, and we computed prediction matrices respectively for each generalization test (Supplementary Fig. 1B). We then calculated cosine similarity between confusion matrices of the monkeys' performance and prediction matrices of the CNN models for each generalization test (Supplementary Fig. 1C), and for pooled data of all generalization tests (Fig. 2D; Supplementary Table 2; Supplementary Fig. 3A). To quantify the significance of the correlations for the pooled data, we employed a permutation test described in three steps. First, we permuted the values of the monkeys' performance confusion matrices 10,000 times, resulting in 10,000 random pairings between the monkeys' performance and the model predictions. Then, we computed the cosine similarity for each permuted pair, yielding a permutation distribution of similarity values. We defined the baseline similarity level as the expected value of this permutation distribution. Finally, the significance ($p$-value) of the observed cosine similarity between the true monkeys' performance confusion matrices and the model predictions was quantified as the proportion of permuted pairs with a cosine similarity higher than the observed value. We set the significance threshold at $p < 0.05$.

We extracted features from internal layers of the three two-stream CNNs. All networks implement a ResNet-50–derived backbone, organized into six successive blocks (maxpool, layer1–4, and the classification head)[63]. For each video, we recorded the activation pattern at the last layer of each block as intermediate feature representations, using the following tap points in the MMACTION2 implementation: backbone.maxpool, backbone.layer1.2.relu, backbone.layer2.3.relu, backbone.layer3.5.relu, backbone.layer4.2.relu, and cls_head.dropout. Activations across two streams were averaged to yield a single feature vector per video and layer for each model. These layer-wise feature vectors were then used as the CNN-derived representations of the stimuli in subsequent analyses linking with monkeys' action categorization patterns. The output of the deepest layer before the final class output from each model was used as the high-level features of the videos.

## Estimation of low-level features

We evaluated two low-level features of our videos, luminance and contrast. Luminance indicates the average level of pixel intensity of an image. We measured luminance for each frame in a video as the average grayscale intensity across all pixels. Contrast indicates the range of pixel intensity of an image. We measured contrast for each frame in a video, as the standard deviation of all pixels' grayscale intensity values.

We used Python-based functions or packages to estimate both low-level and dynamic spatial and temporal features (see below). For low-level features, we computed features of luminance and contrast with custom-made scripts.

Feature models had a sampling frequency of 30 Hz, matching the stimulus display in the experiment. Video frames were gray-scaled and down-sampled to $180 \times 160$ pixels size for efficient computation.

## Estimation of dynamic spatial and temporal features

Mask definition. To estimate the spatial-temporal features, we accounted for the major action components of our videos. We employed 5 sets of action-related body parts as masks: hand, hand-and-arm, agent, agent without hand, and agent without hand and arm. We segmented single hand, arm with hand, and whole agent (Fig. 3A) from the action videos using Detectron2[64]. Specifically, we used the Detectron2 R50-FPN baseline model fine-tuned on 300 frames of our stimuli, annotated using the VIA Annotation Software[65] and Segment-Anything[66].

Dynamic spatial features. We extracted five computationally efficient and conceptually interpretable spatial features (based on single frames) from our stimuli, including HOG, edges, silhouette, silhouette size, and Hu moments of the silhouette (Fig. 3B, Supplementary Video 1). HOG describes the local spatial structure of a visual stimulus, capturing information of object edges and textures, robust to small changes in pose, appearance, and illumination. We used the hog function from the open-source image processing library Scikit-Image[67] to compute HOG. The algorithm first computes image gradients using the Sobel-Feldman operator, then summarizes gradients within cells of $8 \times 8$ pixels size into 9 orientation bins equally spanning 180°, weighted by gradient magnitude, and normalized across blocks of $2 \times 2$ cells size. For each set of stimulus segmentation (see Mask definition), blocks located within the segmentation mask were filtered, and the 9-bin histogram of each block, along with the block center position were used as the HOG feature of the specific segmentation. Edges capture discontinuities in pixel intensity in an image, typically corresponding to object boundaries. Edges were computed using OpenCV-Python's Canny edge detector. The stimulus frame was first smoothed by a Gaussian filter with a $3 \times 3$ pixel window size. From the smoothed stimulus frame, edges were detected using a Canny edge detector with [50,200] pixel intensity thresholds, and filtered by a segmentation mask to obtain the edges feature of the specific segmentation. Silhouette and silhouette size were defined as the binary segmentation mask and the sum of the binary mask, respectively. Hu moments are a set of seven descriptors that characterize the shape of an object in an image. This shape estimation is invariant to translation, scale, and rotation, and the first 6 descriptors are also invariant to reflection. We computed Hu moments of every silhouette using OpenCV-Python's moments and HuMoments functions.

Dynamic temporal features include optical flow, histogram of oriented optical flow (HOF), motion energy, movement direction, and movement velocity (Fig. 3C, Supplementary Video 1). Optical flow captures apparent motion in video frames. It provides 2D displacement vectors for all pixels, indicating how they move from one frame to the next. We estimated optical flow using the state-of-the-art method FlowFormer[68] trained on the Sintel dataset. For each segmentation, optical flow vectors of pixels within segmentation masks are extracted as the feature vectors. Similar to HOG, HOF describes local patterns of optical flow vectors based on their orientations and magnitudes[69]. We computed HOF using a customized script, which summarized optical flow vectors, in the same way as HOG, into 9-number feature vectors. Motion energy describes the amount or intensity of motion of all pixels in video sequences. We computed motion energy as the L2 norm of the optical flow vector. Movement direction and velocity describe the motion of silhouettes in video sequences. We first computed the center of a silhouette as the average of all pixels' coordinates, then calculated direction and velocity from the change in silhouette center across frames.

## Correlation between behavioral performance and stimulus features

Behavioral performance and stimulus features vary in scale and dimensionality. To assess the similarity or dissimilarity of these two perspectives,

we first calculated the distance matrices among the action categories for each generalization test, separately for behavioral performance and stimulus features. A distance matrix represents a structured set of differences between the conditions, i.e., [grasping vs touching, grasping vs reaching, touching vs reaching]. This allows heterogeneous datasets to be compared within a common metric space for further analysis.

For behavioral performance, we computed Youden's J statistic[70], which considers both the sensitivity and specificity of the classification performance. Sensitivity is defined as the true positives divided by the sum of true positives and false negatives. Specificity is defined as true negatives divided by the sum of true negatives and false positives. Youden's J statistic is defined as the sum of sensitivity and specificity minus one. Note that before this computation, we adjusted the confusion matrices of behavioral performance according to the performance patterns to account for monkeys' behavioral bias (see "Methods"—Behavioral data analysis for action categorization tasks). Specifically, for the confusion matrices of pattern G-TR, since only touching and reaching actions revealed unclear discrimination, we adjusted the selection ratio of upper and right target for the two action types to 50%; for pattern GTR which indicates failure of the tests, we adjusted the confusion matrices to 33.33% (chance level) for all category-target pairs.

For stimulus features, we treated features of 3D video data as multi-dimensional point sets and described feature patterns across videos by computing the chamfer distance of pairs of videos for each feature[26–28]. The formula shown in Fig. 3E represents the chamfer distance between two feature point cloud sets, M and N. It is computed as follows: for each point in point set M, find the closest point in point set N, compute the squared Euclidean distance, and average over all points in point set M; Vice versa, for each point in point set N, find the closest point in point set M, compute the squared Euclidean distance, and average over all points in point set N. The final chamfer distance is the sum of these two terms, ensuring a bidirectional measure of similarity. All dimensions of a feature contribute equally to the chamfer distance of two point sets. This approach provides an advantage over common methods by preserving the spatial and temporal structure of dynamic features. It retains all information from all dimensions without manipulations such as vectorizing features or reducing one or more dimensions of dynamic features. For efficient computation of chamfer distance between large point sets, we used symbolic tensor operations from the KeOps library[58]. We leveraged the KeOps toolkit with GPU integrations to accelerate distance metrics computation by representing distance matrices in symbolic tensors, using mathematical formulas to define each entry, and computing individual entries during the reduction operation without constructing full matrices. We applied this approach to all stimulus features, including low-level features, dynamic spatial and temporal features, and the high-level features derived from the deepest layers of the two-stream CNNs.

We next computed the correlation between the distance matrices of behavioral performance and stimulus features using cosine similarity. For behavioral performance and each stimulus feature, we pooled distance matrices from all generalization tests, respectively, for behavior performance and each feature. We applied cosine similarity tests between pooled distance matrices of behavior performance and each of the feature distance matrices for individual monkeys and at the group level. To quantify the significance of the correlations, we employed a permutation test described in three steps. First, we permuted the behavioral performance index values 10,000 times, resulting in 10,000 random pairings between the behavioral performance and the stimulus features. Then, we computed the cosine similarity for each permuted pair, yielding a permutation distribution of similarity values. We defined the baseline similarity level as the expected value of this permutation distribution. Finally, the significance ($p$-value) of the observed cosine similarity between the true behavioral performance and the stimulus features was quantified as the proportion of permuted pairs with a cosine similarity higher than the observed value. We set the significance threshold at $p < 0.05$.

## Reporting summary

Further information on research design is available in the Nature Portfolio Reporting Summary linked to this article.

## Data availability

The data that support the findings of this study are partially available online on Zenodo: https://doi.org/10.5281/zenodo.18978813. Full data will be available from the corresponding author upon reasonable request.

## Code availability

The code that supports the findings of this study is available online on Zenodo: https://doi.org/10.5281/zenodo.18978813.

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

## Acknowledgements
We thank W. Depuydt, M. De Paep, S. Kumar, M. Vissers, S. Sharma, P.A. Fiave, L. Sypre, T. Yao, C. Fransen, A. Hermans, P. Kayenbergh, G. Meulemans, I. Puttemans, C. Ulens, M. Verbeeck, and S. Verstraeten for technical and administrative assistance. Funding: This work was supported by Fonds Wetenschappelijk Onderzoek Vlaanderen (G.0.622.08; G.0.593.09; G.0.854.19; 12AHE24N; V475523N) and KU Leuven (C14/17/109; C14/21/111).

## Author contributions
Q.H.J. contributed conceptualization, methodology, formal analysis, visualization, writing/reviewing/editing manuscripts; D.C. contributed conceptualization, methodology, investigation, formal analysis, visualization, writing/reviewing/editing manuscripts, supervision, funding acquisition; K.N. contributed conceptualization, methodology, writing/reviewing/editing manuscripts, supervision, funding acquisition.

## Competing interests
The authors declare no competing interests.

## Additional information

Consent statementConsent was obtained from all identifiable individuals shown in manuscript figures for publication of their images.

