## [Transparent Peer Review file · Communications Biology]

Dynamic Spatiotemporal Features in Action Recognition: A Multimodal Study

Corresponding Author: Dr Ding Cui

Version 0:

Reviewer comments:

Reviewer #1

(Remarks to the Author)

Jin et al. investigated how representations of spatial and temporal features contribute to action recognition, using fMRI and behavioral experiments in monkeys. Three different actions were inspected, including grasping, touching, and reaching, using videos of monkey model or human actors. They found that most brain areas in the action observation network (AON) had distinct representation of grasping versus the other two actions. The authors then aligned the monkeys' behavioral performance with dynamic spatial and temporal features derived from videos using action recognition models, revealing hierarchical and selective correlations between these features and action recognition. The paper is well-written, and its findings—particularly the use of dynamic spatiotemporal features—offer valuable insights into action perception.

Yet the manuscript in its present form has several notable limitations.

The attempt to disseminate the contributions of different spatiotemporal features to action recognition and their corresponding neural representations is compelling. However, the correspondence between the brain representations and the behavioral-model results is difficult to interpret. The absence of a direct alignment between neural data and the different features weakens the strength of the evidence. Demonstrating how these different levels of features are represented in the brain would significantly strengthen the manuscript's conclusions.

The authors showed action-decoding results across a wide range of brain areas, highlighting their differential levels of engagement in action processing. However, early visual areas V1-V4 exhibited the highest decoding performance, substantially higher than the AON regions. Does this implicate that low-level visual features contribute more to action recognition or the results were driven by features that early visual areas are tuned to? The authors should further interpret and clarify these results.

Is it expected that the behavioral performance would align closely with representations from the output layer of the action recognition models, given that these models were transferred-learned using the same videos stimuli? In other words, both the monkeys and the models were trained using the same videos and tasked with classifying the same actions. Isn't it therefore predictable that the two systems would show similar overall performance or similar difference in performance across the three action types?

Additionally, the rationale for selecting the three action types (grasping, touching, and reaching) is unclear. While the introduction provides a solid background on the broader research question, it does not explain how these specific action types capture the complexity of action features. More justification is needed.

Fig. 4C showed that luminance and contrast were not significantly correlated with behavioral performance. What about other low-level features like orientation? Do earlier layers of the action models encode these low-level features? Including a behavior-model correlation across CNN layers may offer a more comprehensive picture of how different levels of action features contribute to action recognition.

Finally, the estimated null distribution should be displayed for Fig. 2D and Fig. 4 to reflect the "baseline" for interpreting the robustness and reliability of reported cosine similarity between behavior and features.

Reviewer #2

(Remarks to the Author)

Jin et al. investigated 1) whether brain representations of action reflect action recognition behavior and 2) how spatiotemporal information contributes to action recognition and the corresponding brain representation. They recorded fMRI while monkeys watched videos of grasping, touching, and reaching actions. They revealed a grasping-dominant activation pattern within the action observation network (AON). Regions within AON can dissociate grasping from the other two actions

but cannot dissociate the other two actions well. This is consistent with monkey's discrimination behavior. Then, They compared categorization confusion matrices between the monkeys and the two-stream CNNs and found a significant correlation. They also extracted spatial and temporal features from action videos and showed that these features, which captured the evolution of information over time, were selectively correlated with behavior, and revealed a complex interplay between features and key action components (hand, arm and body).

This study on action recognition is significant as it provides empirical evidence to fill an important gap in the existing literature and enhance our understanding of the neural and computational mechanisms underlying action recognition. The experiment design is solid and the employment of two-stream CNNs for comparison with monkey behavior is innovative.

Comments:

1. In page 7, the authors claim that monkeys might use similar strategies to two-stream CNNs based on the high correlation between the categorization confusion matrices of monkeys and CNNs. However, this is sort of unconvincing to me considering different subjects could employ distinct features and strategies to achieve the same categorization behavior. Yet, the authors also show that high-level features extracted from CNNs are significantly correlated with monkey behavior (in page 11). Combining these two aspects to suggest that monkeys might use similar strategies to CNNs could be more persuasive.

2. The authors claim that features are utilized in a hierarchical processing framework. However, it is not clear to me what specific results they base this conclusion on as it seems they didn't do the decoding or correlation analysis between features and brain regions?

3. Monkeys can distinguish grasping well in terms of both brain responses and behavior, but have difficulty distinguishing between touching and reaching (similar for CNNs). I wonder whether this is merely due to the closer distances of relevant features between touching and reaching videos, while the distances of relevant features between grasping and the other two actions are larger.

Reviewer #3

(Remarks to the Author)

This study investigates the neural, behavioral, and computational underpinnings of action recognition in macaques, focusing on three types of hand-object interactions: grasping, touching, and reaching. The authors combine fMRI, behavioral categorization tasks, and deep neural network (DNN) models to examine how dynamic spatiotemporal features shape action representations. While the overall framework is ambitious and cross-modal in scope, several methodological and interpretational issues limit the strength and generalizability of the conclusions.

Major Comments:

Visual similarity between "reaching" and "touching" may undermine the core claim of "grasping-dominance"
A central claim of the paper is that both neural and behavioral responses are biased toward grasping. However, the reaching and touching conditions are visually highly similar, differing only in whether the hand makes contact. This subtle distinction is difficult even for human observers to discern in the provided stimuli (at least from the uploaded movie), raising the possibility that the apparent "grasping-dominance" simply reflects the fact that grasping is the only visually distinctive condition. Without a more thorough analysis of stimulus similarity—e.g., in low- and mid-level visual feature space—it is difficult to interpret the observed neural and behavioral asymmetry as cognitively meaningful.

Behavioral biases may reflect task design constraints rather than action understanding

The behavioral paradigm maps each action category to a fixed spatial response (e.g., grasping = left, touching or reaching = up). In the test sessions, monkeys appeared to exhibit strong spatial biases (e.g., frequent choice of the "up" and "left" target). This suggests that their choices may be influenced by motor habits or task-specific heuristics, rather than genuine perceptual discrimination. Given that behavioral data is used to support claims about action representation, the absence of control for such biases significantly weakens the behavioral evidence.

A stimulus set of only three actions is insufficient for robust model comparisons

The use of only three categories—two of which are hard to distinguish—severely limits the interpretability of comparisons between brain, behavior, and DNNs. The reported similarity between monkey behavior and model predictions may simply reflect shared detection of obvious features (e.g., grasp closure) rather than meaningful alignment of internal representations. To make strong claims about model-brain correspondence or hierarchical encoding principles, a broader and more diverse set of action stimuli is essential.

Version 1:

Reviewer comments:

Reviewer #1

(Remarks to the Author)

I thank the authors for addressing my questions and appreciate the clarifications and additional analyses. However, my concern regarding whether the behavior-model similarity is driven primarily by low-level visual features rather than action-

related features remains unresolved. The layer-wise results indicate that early-layer features are also correlated with behavior (Supplementary Fig. 2), in some cases even more strongly than later-layer features. Taken together with the neural findings showing higher decoding performance in early visual areas (V1–V4), this suggests that the observed behavior–model correspondence may be largely driven by low-level visual properties of the stimuli. Consequently, the above-chance decoding performance observed in the AON could potentially reflect visual gain rather than action-specific representations. Addressing this issue would likely require either more tightly controlled stimuli or a substantially larger stimuli set. In addition, it would be helpful if the authors could further clarify how the three models capture distinct aspects of action recognition. The layer-wise results appear to vary substantially across models, making it difficult to interpret the specific contributions of each.

Finally, I found it very difficult to interpret the results presented solely in tables without accompanying graphical visualization. A simplified plot—for example, line plots with shaded error bars—would greatly improve readability. Plotting three lines across ROIs, with dashed lines indicating the permutation-based null distribution, might effectively convey the key results.

Reviewer #2

(Remarks to the Author)

I would like to thank the authors for their detailed responses to my comments. The authors have addressed the majority of my concerns, and their revisions have significantly improved my understanding of the results and made the manuscript more rigorous and persuasive.

I have only one remaining minor reflection regarding Point #3. My understanding is that "touching" and "reaching" are visually closer on the authors-defined continuum, which leads to the monkeys' difficulty in distinguishing between these two actions (Fig. 2B). I agree that this near-boundary contrast is helpful for demonstrating that low-level features and early visual regions do not directly contribute to the action categorization process. However, I believe this still complicates the interpretation of the AON regions. To take an extreme example: if the monkeys do not distinguish between "touching" and "reaching" at all, they might perceive them both as a single "Action A," while perceiving "grasping" as "Action B." In this scenario, the task essentially collapses into a binary classification (A vs. B) for the monkey, where we can not define "A-dominance" or "B-dominance" in AON regions.

But my concern here is limited. I believe the study as a whole is strong and I am somewhat hesitant. I also note that Reviewer #3 raised a similar point. Therefore, I recommend the manuscript for publication and leave it to the Editor to weigh our collective feedback on this point.

Reviewer #3

(Remarks to the Author)

The authors have addressed all of my concerns.

Version 2:

Reviewer comments:

Reviewer #1

(Remarks to the Author)

Thank the authors again for addressing my remaining questions.

Point-to-point reply:

We would like to thank all three reviewers for their constructive comments. We have responded to each of their comments and revised the manuscript in response to these comments. We have performed further analyses to provide additional evidence to address some of the reviewers' concerns. We revised figures and added additional tables and texts/paragraphs in different sections providing additional information/clarity for our claims. Below, you can find our responses to the reviewer' comments and changes we made to the manuscript.

Reviewer #1 (Remarks to the Author):

Jin et al. investigated how representations of spatial and temporal features contribute to action recognition, using fMRI and behavioral experiments in monkeys. Three different actions were inspected, including grasping, touching, and reaching, using videos of monkey model or human actors. They found that most brain areas in the action observation network (AON) had distinct representation of grasping versus the other two actions. The authors then aligned the monkeys' behavioral performance with dynamic spatial and temporal features derived from videos using action recognition models, revealing hierarchical and selective correlations between these features and action recognition. The paper is well-written, and its findings—particularly the use of dynamic spatiotemporal features—offer valuable insights into action perception.

Yet the manuscript in its present form has several notable limitations.

The attempt to disseminate the contributions of different spatiotemporal features to action recognition and their corresponding neural representations is compelling. However, the correspondence between the brain representations and the behavioral-model results is difficult to interpret. The absence of a direct alignment between neural data and the different features weakens the strength of the evidence. Demonstrating how these different levels of features are represented in the brain would significantly strengthen the manuscript's conclusions.

Response:

We appreciate this point. Our study was designed in two phases for methodological reasons. First, we mapped brain responses during naïve, passive observation to avoid training-induced plasticity that can reshape action representations (see Cui et al., 2023). Only afterward did we train behavioral categorization with a minimal number of examples to both establish behavioral readouts and preserve most videos for generalization tests. Expanding the stimulus set would have markedly lengthened the time required for monkey training and testing, so we balanced sample size against feasible experimental duration in advance. Using the *same* extensive stimulus set for both passive fMRI and trained

categorization would confound interpretations because training would alter cortical responses.

Within these constraints, we provide converging evidence linking behavior, features, and brain: (1) Behavioral categorization of the fMRI stimuli showed a grasping-dominant response pattern (Fig. 2B), and this pattern generalizes to additional videos (Fig. 2C); (2) Low-level pixel summaries (luminance/contrast) do not track behavior (Fig. 4C), whereas dynamic spatiotemporal features and CNN layers correlate with behavior (Fig. 4; Suppl. Fig. 2), indicating distributed recruitment across a feature hierarchy; (3) Brain patterns show that early visual areas (V1-V4) express sensitivity to low-level/retinotopic differences and do not mirror behavior, whereas AON regions express the action-selective pattern that does parallel behavior (Fig. 1). We present the brain results as supportive (but indirect) evidence consistent with the feature-behavior findings.

We agree that a direct feature-brain mapping would strengthen the manuscript. We now add it to Limitations, noting that feature-brain encoding/decoding (e.g., voxel-wise encoding models, condition-rich RSA) with a larger, more varied action set is a planned next step (ongoing work in our lab). Our goal here was to establish, under a clean and controlled design, which information relates to behavior; a full feature-brain mapping will follow in a dedicated study.

In p.13, line 30-32, we added: “In parallel, direct feature-brain mapping (e.g. voxel-wise encoding models or condition-rich RSA with a larger, more varied action set) will be a natural next step to fully characterize representational organization of our dynamic features in the primate brain.”

The authors showed action-decoding results across a wide range of brain areas, highlighting their differential levels of engagement in action processing. However, early visual areas V1-V4 exhibited the highest decoding performance, substantially higher than the AON regions. Does this implicate that low-level visual features contribute more to action recognition or the results were driven by features that early visual areas are tuned to? The authors should further interpret and clarify these results.

Response:

We agree that V1–V4 show the highest decoding accuracies in our MVPA results. We interpret this result as sensitivity to low-level and positional stimulus differences (e.g., luminance, contrast, local motion, retinotopic layout) that naturally vary across our action videos but not as evidence that early visual cortex contributes more to action recognition per se. Two clarifications separate stimulus discriminability from action identity.

First, decoding \neq category readout. MVPA quantifies how well voxel patterns separate conditions; it will be sensitive to any stable visual differences. Because early visual areas are

tuned to local pixel-level features, high decoding accuracies in V1-V4 are expected even when those features are not the ones the monkeys use to assign action categories.

In addition, despite lower absolute accuracies than V1-V4, most AON ROIs exhibit the grasping-dominant dissociation (grasping vs touching/reaching) that mirrors the monkeys' behavior, indicating that this network carries feature-integrative, action-selective information. In contrast, simple low-level summaries (luminance/contrast) do not track behavior (Fig. 4C), showing that trivial low-level cues are insufficient to explain the behavioral pattern.

We have clarified this distinction in the manuscript.

In p.12, line 25-27, we added: "In our results, high decoding accuracies in V1–V4 are expected from sensitivity to retinotopic layout and low-level variations that differ across the three action categories."

In p.12, line 31-33, we revised: "Consistent with this view, prior work shows that behaviorally relevant features better track AON representations than generic labels or low-level summaries⁴⁰."

Is it expected that the behavioral performance would align closely with representations from the output layer of the action recognition models, given that these models were transferred-learned using the same videos stimuli? In other words, both the monkeys and the models were trained using the same videos and tasked with classifying the same actions. Isn't it therefore predictable that the two systems would show similar overall performance or similar difference in performance across the three action types

Response:

Our goal in matching train/test materials was to eliminate trivial stimulus-set differences so that any potentially observed distinct categorizations speak to processing principles rather than to mismatched inputs. We agree that using the same stimulus sets for monkeys and models might increase comparability. However, matching stimuli does not guarantee similar performance or confusion structure. Two learners exposed to the same train/test videos can diverge markedly unless they share task-independent constraints that shape generalization.

Three points support this interpretation:

- Our two-stream networks were heavily pretrained on large action datasets and only lightly fine-tuned with early-epoch selection to avoid overfitting in our task (Methods, p.19–20). Their representational geometry is therefore shaped primarily by broad spatiotemporal statistics, not by the small set of task stimuli. Conversely, the monkeys had daily-life action exposure but no prior action-categorization training

and only brief training with our 12 training videos in the current study. Given these distinct learning histories and architectures, shared train/test stimuli in our task alone do not dictate similar generalization or confusions.

- It's the 'pattern' we compare, not exact accuracy. We correlate behavioral confusion structure with model prediction structure. Even when overall difficulty is shared (same stimuli), the off-diagonal confusions would not be the same. In practice, different algorithms trained on the same videos often produce distinct confusion patterns (Suppl. Fig. 1); alignment here is thus informative rather than inevitable.
- For the CNNs we used, we deliberately crossed architecture (I3D and TSN) and pretraining dataset (K400 and SSV2) revealing three different CNN models: I3D-K400; TSN-K400; TSN-SSV2. All three variants, particularly the two TSN models (same architecture with different datasets), showed similar alignment with monkeys (Fig. 2D; Suppl. Fig. 1). This pattern indicates that the effect is robust across architectures and pretraining datasets, and is not a trivial artifact of using shared stimulus sets.

For further clarity, we edited the text in the Discussion.

In p.11, line 39-40, we revised: "We observed significant correlations between monkeys' performance patterns and that of two-stream CNN models (Fig. 2D). "

In p.12, line 6-8, we revised: "Taken together, the robustness of the alignment across model families and the observed feature hierarchy support the interpretation of convergent solutions under shared task constraints in how spatial and temporal information are leveraged during action processing."

Additionally, the rationale for selecting the three action types (grasping, touching, and reaching) is unclear. While the introduction provides a solid background on the broader research question, it does not explain how these specific action types capture the complexity of action features. More justification is needed.

Response:

Our three actions were selected to sample a graded contact/closure dimension of hand-object interaction under tight control of nuisance factors. Specifically, reaching → touching → grasping implements a no-contact → contact → enclosure continuum while holding constant the actor, object, viewpoint, background, lighting, and approach kinematics. This design served our two goals: 1) examining the extent to which spatial-temporal features are recruited to support behavior; and 2) probing whether brain representations track this graded continuum in ways that predict behavioral categorizations (thus linking features to brain via behavior).

By varying interaction state without changing scenes, we can attribute behavioral differences to measurable dynamic features (e.g., HOG/edges, optical flow/HOF, silhouettes, Hu moments) rather than uncontrolled factors, yielding a mapping from feature sets to behavioral confusions. Second, using the same tightly controlled actions for monkeys and models removes trivial stimulus-set confounds, so any alignment in performance structure reflects shared weighting of dynamic cues, not mismatched inputs. Third, a compact set kept monkey training and fMRI sampling tractable while still enabling generalization to unseen samples and replication across architectures (I3D-K400, TSN-K400, TSN-SSV2).

In Introduction, we added the rationale for selecting the three action types (p.3, line 5-7): “We selected this trio to implement a graded contact/closure continuum of hand-object interaction, i.e., no contact → contact → enclosure, while holding actor, object, viewpoint, background, lighting, and approach kinematics constant.”

In Methods (p.14, line 37-42), we added: “We selected reaching, touching, and grasping to vary contact and enclosure along a graded continuum while holding approach kinematics and scene layout constant (same actor, object, viewpoint, background, lighting). This controlled manipulation isolates effector-object interaction while minimizing unrelated variance, enabling targeted analysis of dynamic features. The compact action set also ensured monkey training and fMRI sampling efficient, with held-out video samples used to assess generalization (see *Action categorization task*).”

We are aware of that three categories do not exhaust action space; accordingly, we added a Limitations/Future Directions paragraph noting that broader, more diverse actions will allow richer tests of model-brain correspondence and representational geometry at scale. Our present design should be viewed as a controlled, mechanistically interpretable slice of action space optimized for feature-behavior analysis and for linking graded interaction state to brain representations that predict behavior.

Fig. 4C showed that luminance and contrast were not significantly correlation with behavioral performance. What about other low-level features like orientation? Do earlier layers of the action models encode these low-level features? Including a behavior-model correlation across CNN layers may offer a more comprehensive picture of how different levels of action features contribute to action recognition.

Response:

We considered luminance and contrast as the most influential low-level, raw-pixel dimensions. We also explicitly included HOG and edge descriptors, both orientation-sensitive, as lower-level/less-integrative features in our dynamic spatial-temporal feature series. In our results, HOG correlated significantly with monkeys’ performance patterns

across all body-segmentation masks, whereas edges showed significant correlations primarily within the hand mask.

To assess the association between CNN layer representations and behavior, we conducted an additional layer-wise analysis. For each model (I3D-K400, TSN-K400, TSN-SSV2), we extracted six hierarchical block features (the last layer of each block), computed feature distances across categories for each block, and correlated the feature distances with monkey behavioral patterns (we added in Methods, p.20, lines 6-17). The results (we added Suppl. Fig. 2, Suppl. Table 5) show: (1) in all three models, the top two blocks (layers 5–6) exhibit significant correlations with behavior (both individual monkeys and group); (2) TSN-SSV2 additionally shows significant correlations at intermediate blocks (layers 3–4); and (3) I3D-K400 reveals significant correlations across all blocks, including layers 1–2. Together with our dynamic-feature findings, these results provide convergent evidence that action recognition draws on spatiotemporal information distributed across the feature hierarchy, a pattern reflected in both the monkeys and the two-stream CNNs used here. We have added the corresponding descriptions to the Results and Discussion.

In Results (p.11, line 18-22), we added: “Finally, layer-wise correlations (Supplementary Fig. 2; Supplementary Table 5) show that: across all three models, the top two layers (5–6) reliably correlate with behavior in both individual monkeys and the group; in TSN-SSV2, significant correlations also appear at intermediate layers (3–4); and in I3D-K400, significant correlations extend across the entire hierarchy, including the lowest layers (1–2).”

In Discussion (p.11, line 46-47, p.12, line 1), we added: “Similarly, behavior-CNN layer correlations (Supplementary Fig. 2; Supplementary Table 5) show that behavior-relevant information is present throughout the CNN hierarchy, indicating distributed recruitment of spatiotemporal features across depth.” In p.12, line 6-8, we added “Taken together, these findings support the interpretation of convergent solutions under shared task constraints between monkeys and the CNNs in how spatial and temporal information are leveraged during action processing.”

Finally, the estimated null distribution should be displayed for Fig. 2D and Fig. 4 to reflect the “baseline” for interpreting the robustness and reliability of reported cosine similarity between behavior and features.

Response:

We agree that showing the null (permutation) baseline improves interpretability. For all cosine-similarity analyses we computed label-permutation nulls (baseline) as follows:

- Fig. 2D: for each monkey × model pair.

- Fig. 4: for each feature family (spatial, temporal, low/high-level) for each monkey and the group.
- Suppl. Fig. 2: for each model × layer for each monkey and the group.

Because these baselines span multiple animals, models, feature families, and layers with different dimensionalities, overlaying them directly on the figures made the plots visually dense and harder to read. Instead, we now provide the full numerical results—observed cosine similarity, permutation-baseline mean, and permutation p-value—in Suppl. Tables 2, 4, and 5 (the former Table 2 is now Table 3). The main text (Methods, Results, Discussions, and figure legends) has been updated to include description of baseline calculations, and to point to these tables. Our conclusions are unchanged; the reported effects remain significant relative to the permutation baseline as documented in the tables.

Reviewer #2 (Remarks to the Author):

Jin et al. investigated 1) whether brain representations of action reflect action recognition behavior and 2) how spatiotemporal information contributes to action recognition and the corresponding brain representation. They recorded fMRI while monkeys watched videos of grasping, touching, and reaching actions. They revealed a grasping-dominant activation pattern within the action observation network (AON). Regions within AON can dissociate grasping from the other two actions but cannot dissociate the other two actions well. This is consistent with monkey's discrimination behavior. Then, They compared categorization confusion matrices between the monkeys and the two-stream CNNs and found a significant correlation. They also extracted spatial and temporal features from action videos and showed that these features, which captured the evolution of information over time, were selectively correlated with behavior, and revealed a complex interplay between features and key action components (hand, arm and body).

This study on action recognition is significant as it provides empirical evidence to fill an important gap in the existing literature and enhance our understanding of the neural and computational mechanisms underlying action recognition. The experiment design is solid and the employment of two-stream CNNs for comparison with monkey behavior is innovative.

Comments:

1. In page 7, the authors claim that monkeys might use similar strategies to two-stream CNNs based on the high correlation between the categorization confusion matrices of monkeys and CNNs. However, this is sort of unconvincing to me considering different subjects could employ distinct features and strategies to achieve the same categorization behavior. Yet, the authors also show that high-level features extracted from CNNs are

significantly correlated with monkey behavior (in page 11). Combining these two aspects to suggest that monkeys might use similar strategies to CNNs could be more persuasive.

Response:

We agree that performance alignment alone does not establish that monkeys and CNNs use the same strategies. To address this, we added a layer-wise analysis spanning all hierarchical blocks of each CNN model (we added details in Methods, p.20, lines 6-17). Briefly here, for each model (I3D-K400, TSN-K400, TSN-SSV2), we extracted six hierarchical block features (the last layer of each block), computed category distances for each block feature (layer), and correlated these features with monkey behavior (we added results in Suppl. Fig. 2, and Suppl. Table 5). Despite model-specific differences, the overall pattern shows significant behavior-feature correlations distributed across the hierarchy, paralleling the pattern we report for our dynamic spatial-temporal features.

In Results (p.7, line 45), we removed “The results suggest that monkeys may utilize similar strategies for processing spatiotemporal information to discriminate the observed actions as the two-stream CNNs.” In Discussion (p.11, lines 46-47, p.12, line1, 6-8), we added “Similarly, behavior-CNN layer correlations (Supplementary Fig. 2; Supplementary Table 5) show that behavior-relevant information is present throughout the CNN hierarchy, indicating distributed recruitment of spatiotemporal features across depth.” and “Taken together, these findings support the interpretation of convergent solutions under shared task constraints between monkeys and the CNNs in how spatial and temporal information are leveraged during action processing.”

2. The authors claim that features are utilized in a hierarchical processing framework. However, it is not clear to me what specific results they base this conclusion on as it seems they didn't do the decoding or correlation analysis between features and brain regions?

Response:

We appreciate the opportunity to clarify our claim. We are not asserting a hierarchical processing framework in the brain (a “how” question). Rather, our conclusion concerns what information relates to behavior (a “what” question): the features associated with behavioral categorization are recruited across a feature hierarchy—from lower-level, less-integrative descriptors (HOG/edges/optical flow) to more integrated shape representations (Hu moments) and high-level CNN layers that capture abstract spatiotemporal structure.

The results shown in Fig. 1 are consistent with a widely reported distinction between processing in early vs higher visual regions, and provide supportive (but indirect) evidence to our feature-behavior findings. Namely, early visual areas (V1–V4) show patterns we interpret as sensitivity to low-level/retinotopic differences and therefore do not mirror behavior,

whereas AON regions (proposed to encode higher level stimulus features) express the action-selective pattern (grasping-dominant) that does parallel behavior. Building on previous studies, we outlined (p.12-13) how our behavior-relevant dynamic features might correspond to representational gradients within the AON, in the context of hierarchical organization in both the feature set and AON subregions. This discussion is intended to situate our findings rather than assert a tested processing hierarchy.

In Discussion (p.12, line 35-38), we revised: “Specifically, building upon the action recognition model proposed by Fleischer et al.² and other previous studies^{10,13,15,41–50}, our results can be interpreted within the context of proposed hierarchical organization in the AON (without asserting a processing hierarchy).”

3. Monkeys can distinguish grasping well in terms of both brain responses and behavior, but have difficulty distinguishing between touching and reaching (similar for CNNs). I wonder whether this is merely due to the closer distances of relevant features between touching and reaching videos, while the distances of relevant features between grasping and the other two actions are larger.

Response:

We selected the three types of actions to form a graded contact/closure continuum of hand-object interaction—reaching (no contact) → touching (contact) → grasping (contact + enclosure)—under matched approach kinematics and a fixed scene layout (same actor, object, viewpoint, background, lighting). We agree that touching and reaching are adjacent on this continuum (contact vs no contact), whereas grasping adds hand-closure/enclosure, which is more diagnostic. Thus, touching vs reaching forms a near-boundary contrast and grasping vs the others a farther-boundary contrast, making greater confusion between touching and reaching expected and informative, not problematic, because it reveals which dynamic cues are weighted when discriminability is low. Under these controls, the increased discriminability of grasping naturally follows from its additional dynamic cues (hand shaping/closure, enclosure), while touching and reaching differ mainly by contact onset. The observed behavioral pattern—and the grasping-dominant dissociation in the AON—therefore reflects our graded contact/closure rationale.

Simple pixel-level summaries (luminance/contrast) do not track behavior (Fig. 1B; Fig. 4C), arguing against an “obvious pixel cue” account. Instead, the alignment across monkeys and two-stream CNNs points to reliance on dynamic spatiotemporal cues (e.g., trajectory near contact, local motion and shape evolution around the effector) that are inherently weaker for distinguishing touching from reaching than for distinguishing grasping.

We clarify in the manuscript that touching and reaching were adjacent to probe feature-based recruitment under low discriminability, and we note in Limitations/Future

Directions that a broader action set will allow more exhaustive tests of model-brain correspondence across independent discriminative axes (e.g., tool use, transitivity, social context, viewpoint/object variation).

In Discussion (p.13, line 21-32), we added: “Our stimulus set samples a graded contact/closure continuum (reaching, touching, grasping) under tightly controlled scene and kinematic factors to maximize interpretability for feature-behavior-brain comparisons. The near-boundary nature of touching vs reaching produces higher confusability than grasping and is informative for our feature-behavior focus because it reveals which dynamic cues are weighted when discriminability is low. While this design isolates diagnostically relevant spatiotemporal information, it may limit inferences about broader action taxonomies. Future work will expand the action space (e.g., transitivity, tool use, social vs nonsocial contexts, viewpoint/object variation) to test whether the patterns we observe generalize across richer repertoires. In parallel, direct feature-brain mapping (e.g. voxel-wise encoding models or condition-rich RSA with a larger, more varied action set) will be a natural next step to fully characterize representational organization of our dynamic features in the primate brain.”

Reviewer #3 (Remarks to the Author):

This study investigates the neural, behavioral, and computational underpinnings of action recognition in macaques, focusing on three types of hand-object interactions: grasping, touching, and reaching. The authors combine fMRI, behavioral categorization tasks, and deep neural network (DNN) models to examine how dynamic spatiotemporal features shape action representations. While the overall framework is ambitious and cross-modal in scope, several methodological and interpretational issues limit the strength and generalizability of the conclusions.

Major Comments:

Visual similarity between “reaching” and “touching” may undermine the core claim of “grasping-dominance”

A central claim of the paper is that both neural and behavioral responses are biased toward grasping. However, the reaching and touching conditions are visually highly similar, differing only in whether the hand makes contact. This subtle distinction is difficult even for human observers to discern in the provided stimuli (at least from the uploaded movie), raising the possibility that the apparent “grasping-dominance” simply reflects the fact that grasping is the only visually distinctive condition. Without a more thorough analysis of stimulus similarity—e.g., in low- and mid-level visual feature space—it is difficult to interpret the observed neural and behavioral asymmetry as cognitively meaningful.

Response:

We appreciate the concern and address it on design rationale, interpretation of decoding, evidence against trivial low-level explanations, and scope.

Our three action types implement a graded contact/closure continuum of hand–object interaction—reaching (no contact) → touching (contact) → grasping (contact + enclosure)—under matched approach kinematics and a fixed scene layout (same actor, object, viewpoint, background, lighting). This design isolates effector-object interaction while minimizing unrelated variance. Consequently, touching vs reaching is a near-boundary contrast (expected to be harder), whereas grasping vs touching/reaching introduces additional diagnostic cues (hand shaping/closure, enclosure). Greater separability for grasping is therefore an interpretable outcome of this continuum rather than an artifact. Moreover, monkeys succeeded in several generalization tests (green/small/ring/right-hand; Suppl. Fig. 1A), indicating that even under the near-boundary touching-reaching regime, our design supports accurate categorization.

By construction, the three actions also differ in lower-level visual properties. This is reflected in high decoding in V1-V4, which we interpret as sensitivity to low-level/positional differences (retinotopic layout, edges/orientation, contrast, luminance, local motion). In contrast, many AON regions display the grasping-dominant dissociation that mirrors behavior, consistent with feature-integrative, action-selective coding. Notably, although touching vs reaching was designed to be more similar, the contact vs no-contact distinction is nevertheless captured: somatosensory area S2 shows significant T–R decoding (Fig. 1B). This dissociation (effector-object non-interactive reaching versus touching) aligns with prior human and macaque fMRI evidence implicating S2 in action-related somatosensory processing (Keysers et al., 2004; Sharma et al., 2018).

Simple pixel-level summaries (luminance/contrast) do not track behavior (Fig. 1B; Fig. 4C), arguing against an explanation based on “obvious” pixel cues. Instead, the pattern is consistent with dynamic spatiotemporal cues (e.g., local motion around the effector, silhouette/shape evolution) that are inherently weaker for distinguishing touching vs reaching than for grasping vs the others. A similar pattern across multiple two-stream CNNs further supports convergent solutions under shared task constraints alongside the monkeys.

We do not claim “grasping-dominance” as a universal rule for action recognition. In our controlled continuum, grasping occupies the high end and is therefore more separable. We use this term descriptively for our data. Notably, this pattern is also consistent with prior evidence in monkey single-unit studies: observed grasping is the most represented hand action, both relative to other hand actions and to other action classes (Lanzilotto et al., 2019, 2020), likely reflecting its ethological importance. Our goal here was to link feature-based recruitment to behavior (and to consistent brain patterns) under tight stimulus control, not to exhaustively taxonomize actions or propose stereotypes of action recognition. We now state this scope explicitly and have added Limitations/Future Directions

noting that broader action repertoires will enable stronger tests across independent dimensions of the action space.

In Discussion (p.13, line21-32), we added: “Our stimulus set samples a graded contact/closure continuum (reaching, touching, grasping) under tightly controlled scene and kinematic factors to maximize interpretability for feature-behavior-brain comparisons. The near-boundary nature of touching vs reaching produces higher confusability than grasping and is informative for our feature-behavior focus because it reveals which dynamic cues are weighted when discriminability is low. While this design isolates diagnostically relevant spatiotemporal information, it may limit inferences about broader action taxonomies. Future work will expand the action space (e.g., transitivity, tool use, social vs nonsocial contexts, viewpoint/object variation) to test whether the patterns we observe generalize across richer repertoires. In parallel, direct feature-brain mapping (e.g. voxel-wise encoding models or condition-rich RSA with a larger, more varied action set) will be a natural next step to fully characterize representational organization of our dynamic features in the primate brain.”

Behavioral biases may reflect task design constraints rather than action understanding

The behavioral paradigm maps each action category to a fixed spatial response (e.g., grasping = left, touching or reaching = up). In the test sessions, monkeys appeared to exhibit strong spatial biases (e.g., frequent choice of the “up” and “left” target), This suggests that their choices may be influenced by motor habits or task-specific heuristics, rather than genuine perceptual discrimination. Given that behavioral data is used to support claims about action representation, the absence of control for such biases significantly weakens the behavioral evidence.

Response:

We agree that fixed stimulus-response mappings might, in principle, introduce spatial choice biases. However, our behavioral results cannot be explained by motor habits.

First, during training only correct categorizations were rewarded. Any fixed spatial preference (e.g., always choosing “up/left”) would drive accuracy toward chance ($\approx 50\%$ when two targets are preferred). Instead, we observed high training accuracies ($>90\%$), which against a motor-habit account. Moreover, unlike human participants, monkeys perform under strong reward motivation. Because a fixed spatial preference reduces the overall probability of reward, the reward-maximizing strategy is to use perceptual information to maximize accuracy—which is exactly what we observed with the high training accuracies.

Second, in the generalization phase, $\sim 90\%$ of trials used the trained videos (maintaining high accuracy and motivation), and $\sim 10\%$ were novel test action videos interleaved in random order. For these novel videos, reward was delivered irrespective of

choice to prevent learning during testing, but the 90%/10% mixture preserved the motivation to categorize correctly overall. Under these conditions, monkeys showed category-dependent performance that was significantly above chance in attribute-controlled tests (e.g., green/small/ring/right-hand; Suppl. Fig. 1A). A strong spatial habit would predict perseveration on the preferred target across categories—which we did not observe.

Third, if spatial bias were driving choices, grasping trials should also collapse toward the preferred target (e.g., selecting “up” about half the time considering up/left both were ‘frequently’ selected). Instead, grasping responses remained mostly correct across all tests. Occasional upper target choices appear at a system level (e.g., in the “flip” test, monkey M1 often chose “up” even for grasping) and are best interpreted as fallbacks under uncertainty, not as the driver of overall performance. This interpretation is consistent with grasping’s robust advantage and with confusions concentrated between touching vs reaching—our near-boundary pair by design. In parallel, if a leftward motor habit were driving behavior, we would expect elevated left choices not mainly for grasping but for touching/reaching as well—but this was not observed.

Taken together, the pattern of above-chance accuracy and category-specific generalization supports the conclusion that monkeys performed genuine action categorization, rather than relying on a motor habit.

A stimulus set of only three actions is insufficient for robust model comparisons

The use of only three categories—two of which are hard to distinguish—severely limits the interpretability of comparisons between brain, behavior, and DNNs. The reported similarity between monkey behavior and model predictions may simply reflect shared detection of obvious features (e.g., grasp closure) rather than meaningful alignment of internal representations. To make strong claims about model–brain correspondence or hierarchical encoding principles, a broader and more diverse set of action stimuli is essential.

Response:

We agree that three categories do not exhaust action space, but as already mentioned, reaching → touching → grasping samples a graded contact/closure continuum under matched approach kinematics and a fixed scene layout. This controlled manipulation isolates effector-object interaction while minimizing unrelated variance, allowing us to address our main goals: (1) test the extent to which spatiotemporal features are recruited to support behavior and (2) probe whether brain representations track this graded interaction state in ways that align with behavioral categorizations.

Regarding the concern that two categories are harder to distinguish: in our study, touching and reaching were near each other on the continuum (contact vs no contact). This near-boundary regime is informative because it reveals which dynamic cues (e.g., trajectory,

local optical flow near the effector, silhouette/shape evolution) are weighted when discriminability is low. Notably, simple low-level summaries (luminance/contrast) did not track behavior (Fig. 4C), arguing against an explanation based solely on “obvious pixel cues”. Moreover, the grasping-dominant dissociation observed in behavior also appears in AON patterns, whereas high decoding in V1-V4 is expected from sensitivity to low-level/positional differences that naturally vary across our action videos.

Because the reviewer’s concern centers on model-brain-behavior comparisons, we emphasize that our aim was not to claim mechanism identity, but to test for convergent solutions under shared tasks. Primarily, the models serve Goal 1: they provide a computational benchmark for which spatiotemporal features—from pixel-level descriptors to integrated motion/shape representations—are weighted to support behavior, and they allow a parallel comparison with monkey behavioral categorizations. Goal 2 (brain-behavior linkage) is addressed directly in our data by asking whether brain patterns reflect the graded contact/closure continuum in ways that align with behavioral categorizations; the models are used here only indirectly, as context to interpret which features are plausibly informative. Using matched stimuli, held-out samples for generalization, and diverse architectures/pretraining histories (I3D-K400, TSN-K400, TSN-SSV2) lets us assess convergent feature weighting across systems, specifically for Goal 1, without asserting identical mechanisms.

Finally, we acknowledge our scope. Three categories provide a controlled, mechanistically interpretable slice of action space for feature-behavior analyses and for linking graded interaction state to brain representations. We added a Limitations/Future Directions paragraph noting that broader sets will enable stronger tests of model-brain correspondence and hierarchical encoding at scale, to be pursued in a dedicated follow-up study beyond the scope of the present work.

In Discussion (p.13, line 21-32), we added: “Our stimulus set samples a graded contact/closure continuum (reaching, touching, grasping) under tightly controlled scene and kinematic factors to maximize interpretability for feature-behavior-brain comparisons. The near-boundary nature of touching vs reaching produces higher confusability than grasping and is informative for our feature-behavior focus because it reveals which dynamic cues are weighted when discriminability is low. While this design isolates diagnostically relevant spatiotemporal information, it may limit inferences about broader action taxonomies. Future work will expand the action space (e.g., transitivity, tool use, social vs nonsocial contexts, viewpoint/object variation) to test whether the patterns we observe generalize across richer repertoires. In parallel, direct feature-brain mapping (e.g. voxel-wise encoding models or condition-rich RSA with a larger, more varied action set) will be a natural next step to fully characterize representational organization of our dynamic features in the primate brain.”

Point-to-point reply:

We thank all three reviewers for their additional comments. We have addressed each point, added new supplementary figures, and revised the manuscript accordingly. Below we provide point-by-point responses and summarize the corresponding changes made to the manuscript.

Reviewer #1 (Remarks to the Author):

I thank the authors for addressing my questions and appreciate the clarifications and additional analyses. However, my concern regarding whether the behavior–model similarity is driven primarily by low-level visual features rather than action-related features remains unresolved. The layer-wise results indicate that early-layer features are also correlated with behavior (Supplementary Fig. 2), in some cases even more strongly than later-layer features. Taken together with the neural findings showing higher decoding performance in early visual areas (V1–V4), this suggests that the observed behavior–model correspondence may be largely driven by low-level visual properties of the stimuli. Consequently, the above-chance decoding performance observed in the AON could potentially reflect visual gain rather than action-specific representations. Addressing this issue would likely require either more tightly controlled stimuli or a substantially larger stimuli set.

Response:

We appreciate the reviewer’s thoughtful follow-up. The core concern is whether the observed behavior–model correspondence is driven primarily by low-level visual properties rather than action-related information. Although our study was not designed to adjudicate the ultimate *origin* of monkey–model similarity, our results nevertheless provide several lines of evidence indicating that a purely low-level account is insufficient.

First, we clarify the aim and role of the CNN analyses. Our primary goals were to test (1) the extent to which spatiotemporal features are recruited to support behavioral categorization, and (2) to what extent brain representations align with behavioral structure. The CNNs served two roles consistent with these aims: (i) to instantiate a two-stream logic of action processing motivated by prior computational work, and (ii) to provide high-level, abstract feature representations that complete our explicitly defined low-to-high feature hierarchy. Importantly, the performance alignment between monkeys and CNNs was an observed result, not a hypothesis we set out to test. This alignment (together with behavior-feature correlations and layer-wise results) therefore serves as convergent evidence for our central claim that action recognition draws on spatiotemporal information distributed across a feature hierarchy, a pattern reflected in both monkeys and two-stream CNNs.

It is critical to clarify what we mean by “low-level features.” In our study, the ‘low-level’ features were operationally defined as raw pixel-wise properties, specifically

luminance and contrast. These features do not correlate with behavior (Fig. 4C). By contrast, our spatiotemporal feature set explicitly spans a hierarchy—from relatively low-level descriptors such as HOG and edges (encoding local orientation and contour information) to higher-level descriptors such as Hu moments and CNN-derived features. Thus, features such as HOG—even when evaluated within action-related masks (e.g., hand, arm)—is still considered low-level within our feature hierarchy. The fact that some lower-level descriptors correlate with behavior does not contradict our conclusions; rather, the absence of behavioral correlations for raw pixel statistics, together with more consistent effects at higher feature levels, argues against a purely low-level visual explanation.

Regarding the layer-wise CNN results, we agree that early layers can correlate with behavior in some models. However, there is currently lack of established one-to-one mapping between specific CNN layers and concrete visual features such as HOG and silhouette—either within our models or more generally across architectures. We therefore do not interpret early-layer correlations as evidence that behavior–model similarity is driven solely by raw visual properties. Instead, the overall pattern—with consistent correlations at higher layers across all three models, alongside model-specific differences reflecting architectural and training differences—supports a view of distributed recruitment across depth, rather than dominance by early visual features alone. If behavior–model correspondence were driven primarily by low-level features, we would expect the strongest and most consistent effects to be confined to early layers across models, which is not what we observe.

Moreover, the neural results further argue against a low-level account of action categorization. While V1–V4 show high decoding accuracy—indicating that all action categories are visually discriminable at the pixel/retinotopic level (a G-T-R pattern)—these regions do not mirror the behavioral structure, which is characterized by a grasping-dominant pattern (a G-TR pattern). In contrast, AON regions express action-selective patterns that parallel behavioral confusions. We therefore interpret high decoding in early visual cortex as reflecting visual discriminability, not decision-relevant or action-specific coding, whereas the alignment between AON representations and behavior points to higher-level integration relevant for categorization.

Finally, our broader conceptual framework does not assume action-specific features that bypass the visual hierarchy. We treat observed actions as visual stimuli composed of multiple levels of visual features, processed hierarchically. This view is consistent with widely accepted models of visual processing and does not preclude the emergence of action-related representations through the integration of visual features at higher levels. Our findings are compatible with such hierarchical visual processing and do not rely on positing action-exclusive features.

We fully agree that disentangling the precise correspondence between concrete visual features, CNN layers, and brain representations would require larger stimulus sets,

more controlled manipulations, and broader model comparisons. We explicitly acknowledge this in the Limitations/Future Directions section added in the previous revision. Addressing this question lies beyond the scope of the present study.

In addition, it would be helpful if the authors could further clarify how the three models capture distinct aspects of action recognition. The layer-wise results appear to vary substantially across models, making it difficult to interpret the specific contributions of each.

Response:

Our three networks differ along two dimensions: architecture (I3D vs TSN) and pretraining history (K400 vs SSV2). I3D uses inflated 3D convolutions, which support spatiotemporal integration throughout the hierarchy and capture fine-grained temporal structure more directly. TSN is designed to model longer-range temporal structure by sampling multiple segments and combining them via a consensus function. Pretraining also matters. We included models pretrained on K400, which spans diverse whole-body human actions, and on SSV2, which emphasizes fine-grained manipulative actions involving hand–object interactions. Differences in pretraining data can shift where discriminative information is concentrated along the hierarchy because the models are optimized under different label sets and visual statistics.

Together, these models serve complementary purposes in our study. I3D-K400 emphasizes dense spatiotemporal integration via 3D convolutions with strong transfer from K400. TSN-K400 implements segment-based temporal aggregation under the same K400 pretraining; thus, differences relative to I3D-K400 primarily reflect architectural differences. TSN-SSV2 uses the same TSN architecture but a pretraining set that stresses effector-driven manipulation; thus, differences relative to TSN-K400 primarily reflect pretraining/task statistics. Because the layer-wise correlation is computed against model-derived representations, and these representations vary with architecture and pretraining, differences in the layer-wise behavior–model correlations across models are expected.

Importantly, the layer-wise analysis (Supplementary Fig. 2; Supplementary Table 5) shows a consistent cross-model effect: the highest blocks (layers 5–6) significantly correlate with behavior in all three models. This convergence supports our use of CNN-derived representations as high-level, integrative feature complements to our explicit low-to-high spatiotemporal feature set (Fig. 4). Additional significant correlations at intermediate or earlier blocks in some models likely reflect their distinct temporal-integration schemes and learned priors. Taken together, the results support our central conclusion that behavior relates to spatiotemporal information that is distributed across the feature hierarchy.

We have added clarifying descriptions of the three CNN models in the Methods. On p.19, line 28–31, we added: “I3D uses inflated 3D convolutions, enabling spatiotemporal

integration throughout the hierarchy and capturing fine-grained temporal structure more directly. TSN is designed to model long-range temporal structure by sampling multiple segments and combining them via a consensus function.” On p.19, line 37–39, we added: “This manipulation allowed us to test whether the performance is robust across different architectures and learned priors, rather than being tied to a single source.”

Finally, I found it very difficult to interpret the results presented solely in tables without accompanying graphical visualization. A simplified plot—for example, line plots with shaded error bars—would greatly improve readability. Plotting three lines across ROIs, with dashed lines indicating the permutation-based null distribution, might effectively convey the key results.

Response:

We have provided two additional supplementary figures (suppl. Fig. 3-4) as graphical visualization of supplementary Tables 2, 4, and 5 for better interpretation of the numerical results.

Reviewer #2 (Remarks to the Author):

I would like to thank the authors for their detailed responses to my comments. The authors have addressed the majority of my concerns, and their revisions have significantly improved my understanding of the results and made the manuscript more rigorous and persuasive.

I have only one remaining minor reflection regarding Point #3. My understanding is that "touching" and "reaching" are visually closer on the authors-defined continuum, which leads to the monkeys' difficulty in distinguishing between these two actions (Fig. 2B). I agree that this near-boundary contrast is helpful for demonstrating that low-level features and early visual regions do not directly contribute to the action categorization process. However, I believe this still complicates the interpretation of the AON regions. To take an extreme example: if the monkeys do not distinguish between "touching" and "reaching" at all, they might perceive them both as a single "Action A," while perceiving "grasping" as "Action B." In this scenario, the task essentially collapses into a binary classification (A vs. B) for the monkey, where we can not define “A-dominance” or “B-dominance” in AON regions.

But my concern here is limited. I believe the study as a whole is strong and I am somewhat hesitant. I also note that Reviewer #3 raised a similar point. Therefore, I recommend the manuscript for publication and leave it to the Editor to weigh our collective feedback on this point.

Response:

We appreciate this additional reflection and the opportunity to clarify our interpretation. We do not claim “grasping-dominance” as a universal rule for action recognition or categorization. Rather, the grasping–touching/reaching (G–TR) pattern is one of several representational patterns observed in our data under the current experimental design. Importantly, our results reveal multiple dissociation patterns across behavioral tests and brain regions, indicating multifaceted action processing. In addition to the G–TR pattern observed in both behavior and fMRI (Fig. 1–2; Supplementary Fig. 1), we observed:

- G–T–R (all separable) patterns in three behavioral generalization tests in both monkeys, as well as in decoding results from bilateral V1–V4, left MT, ML, and AIP;
- GTR (non-differentiable) patterns in one behavioral generalization test in both monkeys and in decoding results from right STPa, left TEr, and S1; and
- A T–R (touching–reaching) discrimination pattern in left S2, consistent with contact vs. no-contact differentiation from a somatosensory perspective.

These findings indicate that touching and reaching can be discriminated in specific behavioral contexts and neural substrates, even though they are positioned as a near-boundary pair on our contact/closure continuum and are therefore more confusable overall.

We use the term “grasping-dominant” specifically to describe the G–TR pattern that emerges most consistently across behavior and AON regions among the multiple patterns observed. This terminology is descriptive of the data and does not imply that AON representations reduce to a binary action code. Rather, it reflects that, under our controlled continuum, grasping occupies the high end of the interaction dimension and is therefore more robustly separable, while touching and reaching show graded, context- and region-dependent representations.

We agree that a more detailed characterization of how AON regions represent action categories—particularly with respect to different feature dimensions—would benefit from a larger and more diverse action set, as well as direct feature–brain and category–brain encoding/decoding analyses (ongoing work in our lab). We have acknowledged this limitation and outlined it as a future direction which we added in our previous revision.

Reviewer #3 (Remarks to the Author):

The authors have addressed all of my concerns.

Response:

We thank the reviewer for the positive assessment.